# GensKer: Generative Spectral Kernel with RKHS Expansion Guarantees

## Abstract

Deep spectral kernels are constructed by hierarchically stacking explicit spectral kernel mappings derived from the Fourier transform of the spectral density function. This family of kernels unifies the expressive power of hierarchical architectures with the ability of the spectral density in revealing essential patterns within data, helping to understand the underlying mechanisms of models. In this paper, we categorize most existing deep spectral kernel models into four classes based on the stationarity of spectral kernels and the compositional structure of their associated mappings. Building on this taxonomy, we rigorously investigate two questions concerning the general characterization of deep spectral kernels: (1) Does the deep spectral kernel retain the reproducing property during the stacking process? (2) In which class can the reproducing kernel Hilbert space (RKHS) induced by the deep spectral kernel expand with increasing depth? Specifically, the behavior of RKHS is related to its associated spectral density function. This means that we can implement the deep spectral kernel by directly resampling from an adaptive spectral density. These insights motivate us to propose the generative spectral kernel framework, which directly learns the adaptive spectral distribution by generative networks. This method, with the single-layer spectral kernel architecture, can: (1) generate an adaptive spectral density and achieve deep spectral kernel performance; (2) circumvent the optimization challenges introduced by multi-layer stacking. Experimental results on the synthetic data and several real-world time series datasets consistently validate our findings.

## 1 Introduction

The spectral kernel, derived from the inverse Fourier transform via Bochner's theorem Bochner (1959) or Yaglom's theorem Yaglom (1987), can naturally realize complex-valued spectral kernel mappings, consequently analyzing the data in the frequency domain. At its core lies the spectral density, which is in one-to-one correspondence with the spectral kernel and can effectively uncover underlying patterns within the data. Based on Bochner's theorem, Rahimi and Recht Rahimi & Recht (2007) introduced the spectral representation of stationary kernels. This seminal work inspired a proliferation of subsequent research, such as sparse spectral kernel Lázaro-Gredilla et al. (2010) and spectral mixture kernel Wilson & Adams (2013). These approaches primarily relied on stationary kernels, which depend solely on the distance $\|x - x'\|$ between data $x$ and $x'$. Therefore, they fail to capture richer information in the feature space beyond the simple similarity. To deal with more complex data patterns and tasks, Remes *et al.* Remes et al. (2017) generalized stationary spectral kernels to non-stationary scenarios by defining the spectral density as a mixture of bivariate Gaussian components. This advancement enables the kernels to capture input-dependent patterns as well as long-range dependencies of data, significantly enhancing their modeling capacity.

Benefiting from the remarkable success of the hierarchical architecture in deep neural networks, researchers have integrated this architecture into spectral kernels, leading to the development of deep spectral kernel learning techniques. By stacking explicit spectral kernel mappings, these methods unify the expressive power of hierarchical architectures with the ability of the spectral density in revealing essential patterns within data, providing a principled perspective for understanding the underlying mechanisms of models. Studies Mehrkanoon et al. (2017); Zhang et al. (2017) preliminarily constructed deep spectral kernels by employing an alternating sequence of random Fourier features and linear projections over multiple layers. Subsequently, this framework was extended to

a deeper architecture Mehrkanoon & Suykens (2018) and a more compact version Xie et al. (2019). Meanwhile, Xue *et al.,* Xue et al. (2019) generalized the deep spectral kernel with stationary kernels to non-stationary kernels based on Yaglom's theorem. This work inspired a considerable body of work focused on constructing deep spectral kernels using non-stationary kernels Li et al. (2020); Xue et al. (2023); Li et al. (2022); Tian et al. (2024), which laid the groundwork for further innovations.

By jointly considering the stationarity (stationary or non-stationary) of spectral kernels and the compositional structure (removal or concatenation)[1] of their associated mappings, existing deep spectral kernel models can be systematically categorized into four classes. Building upon this categorization, we present a rigorous investigation of two questions concerning the general characterization of deep spectral kernels: (1) Does the deep spectral kernel retain the reproducing property during the stacking process? (2) In which class can the RKHS of the deep spectral kernel expand with increasing depth? Analysis results reveal that all classes of deep spectral kernels preserve the reproducing property. Furthermore, when the corresponding mapping is compositionally structured by concatenating the real and imaginary components, the induced RKHS is indeed expanding as the number of layers increases. Remarkably, the progressive expansion of RKHS induced by the deep spectral kernels is intrinsically related to its associated spectral density function. This implies that the deep spectral kernel can be implemented by resampling from an adaptive spectral density function. Motivated by this insight, we propose a novel deep spectral kernel learning framework, termed **gen**erative **s**pectral **ker**nel (GensKer), which learns an adaptive spectral density via a generative network. This method possesses two principal advantages: (1) it is capable of generating an adaptive spectral density and thus attaining deep spectral kernel performance using a single-layer spectral kernel architecture. (2) This architecture enables the proposal to circumvent the optimization challenges induced by stacked periodic functions in deep spectral kernels. Our contributions are:

- We present a rigorous investigation central to the reproducing property of deep spectral kernels and the progressive expansion of their associated RKHSs.

- We introduce a novel deep spectral kernel framework, which directly samples from a generated adaptive spectral density to implement the construction of the deep spectral kernel.

- We perform systematic experiments on synthetic data and six real-world time series datasets to verify our findings and proposed method.

## 2 PRELIMINARY

### 2.1 SPECTRAL KERNEL

Spectral kernels, constructed from the inverse Fourier transform, can naturally realize complex-valued spectral kernel mappings, consequently analyzing the data in the frequency domain. These kernels can be broadly classified into two categories: (1) stationary spectral kernels, formulated from Bochner's theorem; and (2) non-stationary spectral kernels, derived from Yaglom's theorem.

**Theorem 1 (Bochner's Theorem)** *Bochner (1959) A stationary kernel $k(\boldsymbol{x}, \boldsymbol{x}') = k(\boldsymbol{x} - \boldsymbol{x}')$ on $\mathbb{R}^d$ is positive definite if and only if it is the Fourier transform of a non-negative measure, such that:*

$$k(\boldsymbol{x} - \boldsymbol{x}') = \int_{\mathbb{R}^d} s(\boldsymbol{\omega}) e^{i\boldsymbol{\omega}(\boldsymbol{x}-\boldsymbol{x}')} d\boldsymbol{\omega}, \quad s(\boldsymbol{\omega}) = \int_{\mathbb{R}^d} k(\boldsymbol{x} - \boldsymbol{x}') e^{-i\boldsymbol{\omega}(\boldsymbol{x}-\boldsymbol{x}')} d(\boldsymbol{x} - \boldsymbol{x}'), \quad (1)$$

*where $s(\boldsymbol{\omega})$ is the spectral density of a non-negative measure.*

**Theorem 2 (Yaglom's Theorem)** *Yaglom (1987) A continuous kernel $k(\boldsymbol{x}, \boldsymbol{x}')$ is positive definite if and only if it admits the following form:*

$$k(\boldsymbol{x}, \boldsymbol{x}') = \int_{\mathbb{R}^d \times \mathbb{R}^d} e^{i(\boldsymbol{\omega}\boldsymbol{x} - \boldsymbol{\omega}'\boldsymbol{x}')} s(\boldsymbol{\omega}, \boldsymbol{\omega}') d\boldsymbol{\omega} d\boldsymbol{\omega}', s(\boldsymbol{\omega}, \boldsymbol{\omega}') = \int_{\mathbb{R}^d \times \mathbb{R}^d} e^{-i(\boldsymbol{\omega}\boldsymbol{x} - \boldsymbol{\omega}'\boldsymbol{x}')} k(\boldsymbol{x}, \boldsymbol{x}') d\boldsymbol{x} d\boldsymbol{x}', \quad (2)$$

*where $s(\boldsymbol{\omega}, \boldsymbol{\omega}')$ is the positive semi-definite bounded variation spectral density of a Lebesgue-Stieltjes measure.*

Theorem 1 and Theorem 2 establish a bijective correspondence between spectral kernels and their spectral densities. This duality reveals that every spectral kernel admits a unique spectral density.

---

[1]The former denotes removing the imaginary component, while the latter represents concatenating the real and imaginary components.

## 2.2 DEEP SPECTRAL KERNEL

**Definition 1 (Deep Spectral Kernel)** *Let $k(\boldsymbol{x}, \boldsymbol{x}') = \mathbb{E}_{\boldsymbol{\omega}}[z_{\boldsymbol{\omega}}(\boldsymbol{x}) z_{\boldsymbol{\omega}}(\boldsymbol{x}')^*] \approx \langle \phi(\boldsymbol{x}), \phi(\boldsymbol{x}') \rangle_{\mathcal{H}}$ be a general kernel, $z_{\boldsymbol{\omega}}(\boldsymbol{x}) z_{\boldsymbol{\omega}}(\boldsymbol{x}')^*$ constitutes an unbiased estimator of $k(\boldsymbol{x}, \boldsymbol{x}')$ under the spectral distribution ($s(\boldsymbol{\omega})$ or $s(\boldsymbol{\omega}, \boldsymbol{\omega}')$). $\phi(\cdot)$ denotes an explicit spectral kernel mapping, and $\mathcal{H}$ is a Hilbert space. The deep spectral kernel with $L$ layers is defined by:*

$$k^L(\boldsymbol{x}, \boldsymbol{x}') = \langle \phi^L(\phi^{L-1} \cdots \phi^1(\boldsymbol{x})), \phi^L(\phi^{L-1} \cdots \phi^1(\boldsymbol{x}')) \rangle. \tag{3}$$

Definition 1 indicates that the core of deep spectral kernels lies in the design of explicit spectral kernel mappings. These mappings are inherently complex-valued and can be obtained by employing the Monte Carlo sampling scheme based on equation 1 and equation 2. To ensure real-valued kernel evaluations, two compositional structures are commonly employed, including removal and concatenation. As a result, four classes of spectral kernel mappings are formulated by:

$$\phi_{s,w/o}(\boldsymbol{x}) = \frac{1}{\sqrt{2M}} \cos(\boldsymbol{\Omega}^\top \boldsymbol{x}), \quad \phi_{ns,w/o}(\boldsymbol{x}) = \frac{1}{\sqrt{4M}} [\cos(\boldsymbol{\Omega}^\top \boldsymbol{x}) + \cos(\boldsymbol{\Omega}'^\top \boldsymbol{x})],$$

$$\phi_{s,c}(\boldsymbol{x}) = \frac{1}{\sqrt{2M}} \begin{bmatrix} \cos(\boldsymbol{\Omega}^\top \boldsymbol{x}) \\ \sin(\boldsymbol{\Omega}^\top \boldsymbol{x}) \end{bmatrix}, \quad \phi_{ns,c}(\boldsymbol{x}) = \frac{1}{\sqrt{4M}} \begin{bmatrix} \cos(\boldsymbol{\Omega}^\top \boldsymbol{x}) + \cos(\boldsymbol{\Omega}'^\top \boldsymbol{x}) \\ \sin(\boldsymbol{\Omega}^\top \boldsymbol{x}) + \sin(\boldsymbol{\Omega}'^\top \boldsymbol{x}) \end{bmatrix}, \tag{4}$$

where $\phi_{s,w/o}(\cdot)$ and $\phi_{s,c}(\cdot)$ denote the stationary spectral kernels in the removal and concatenation scenarios, respectively. $\boldsymbol{\Omega} = [\boldsymbol{\omega}_1, \boldsymbol{\omega}_2, \cdots, \boldsymbol{\omega}_M]$ denotes the frequency matrix, where $\{\boldsymbol{\omega}_i\}_{i=1}^M$ are identically and independently distributed (i.i.d.) and are sampled from $s(\boldsymbol{\omega})$. Similarly, $\phi_{ns,w/o}(\cdot)$ and $\phi_{ns,c}(\cdot)$ correspond to non-stationary spectral kernels, whose frequency pairs $(\boldsymbol{\omega}_i, \boldsymbol{\omega}'i)i = 1^M$ are drawn i.i.d. from $s(\boldsymbol{\omega}, \boldsymbol{\omega}')$ to form the frequency matrices $\boldsymbol{\Omega}$ and $\boldsymbol{\Omega}'$. $M$ is the sampling number.

## 3 THEORETICAL ANALYSIS OF THE TWO QUESTIONS

### 3.1 REPRODUCING PROPERTY OF DEEP SPECTRAL KERNELS

The condition for preserving the reproducing property of functions was initially explored as early as 1995 in FitzGerald et al. (1995). The classical result is presented in Lemma 1.

**Lemma 1** *FitzGerald et al. (1995) Let $g(\cdot)$ be a function on $\mathbb{C}$. Then for any reproducing kernel $k$, $g(k)$ remains a reproducing kernel if and only if $g(\cdot)$ is holomorphic on $\mathbb{C}$ and all the coefficients in its Maclaurin series are nonnegative.*

Following the formulation in Lemma 1, we formally define the stacking of deep spectral kernels as:

$$k^L(\boldsymbol{x}, \boldsymbol{x}') = g(k^{L-1}(\boldsymbol{x}, \boldsymbol{x}')), L \geq 2. \tag{5}$$

The analysis then reduces to specifying the function $g(\cdot)$ for each of the four classes of deep spectral kernels and subsequently verifying their validity under the conditions of Lemma 1. The results are presented in the following Proposition 1 and Proposition 2.

**Proposition 1** *For stationary spectral kernels in the removal and concatenation scenarios, their associated spectral kernel mappings are defined by: $\phi_{s,w/o}(\boldsymbol{x}) = [\cos(\boldsymbol{\omega}_1^\top \boldsymbol{x}), \cdots, \cos(\boldsymbol{\omega}_M^\top \boldsymbol{x})]^\top$ and $\phi_{s,c}(\boldsymbol{x}) = [\cos(\boldsymbol{\omega}_1^\top \boldsymbol{x}), \cdots, \cos(\boldsymbol{\omega}_M^\top \boldsymbol{x}), \sin(\boldsymbol{\omega}_1^\top \boldsymbol{x}), \cdots, \sin(\boldsymbol{\omega}_M^\top \boldsymbol{x})]^\top$, respectively, and $\{\boldsymbol{\omega}_i\}_{i=1}^M \sim \mathcal{N}(\boldsymbol{0}, \sigma^2 \boldsymbol{I})$. The corresponding deep spectral kernels $k_{s,w/o}^L(\boldsymbol{x}, \boldsymbol{x}')$ and $k_{s,c}^L(\boldsymbol{x}, \boldsymbol{x}')$ is formulated as:*

$$k_{s,w/o}^L(\boldsymbol{x}, \boldsymbol{x}') = C_{s,w/o}^{L-1} \cosh(\sigma^2(k_{s,w/o}^{L-1}(\boldsymbol{x}, \boldsymbol{x}'))), L \geq 2,$$

$$C_{s,w/o}^{L-1} = \frac{1}{2} [e^{-\frac{\|\phi_{s,w/o}^{L-1} \cdots \phi_{s,w/o}^1(\boldsymbol{x})\|_2^2 + \|\phi_{s,w/o}^{L-1} \cdots \phi_{s,w/o}^1(\boldsymbol{x}')\|_2^2}{2} \sigma^2}], \tag{6}$$

*and*

$$k_{s,c}^L(\boldsymbol{x}, \boldsymbol{x}') = C_{s,c}^{L-1} \exp(\sigma^2(k_{s,c}^{L-1}(\boldsymbol{x}, \boldsymbol{x}'))), L \geq 2,$$

$$C_{s,c}^{L-1} = \frac{1}{2} [e^{-\frac{\|\phi_{s,c}^{L-1} \cdots \phi_{s,c}^1(\boldsymbol{x})\|_2^2 + \|\phi_{s,c}^{L-1} \cdots \phi_{s,c}^1(\boldsymbol{x}')\|_2^2}{2} \sigma^2}]. \tag{7}$$

**Proof 1** *The proof is relegated to the appendix of our paper due to space limitations.*

**Proposition 2** *For non-stationary spectral kernels in the removal and concatenation scenarios, their associated spectral kernel mappings are defined by $\phi_{ns,w/o}(\boldsymbol{x}) = [\cos(\boldsymbol{\omega}_1^\top \boldsymbol{x}) + \cos(\boldsymbol{\omega}_1'^\top \boldsymbol{x}), \cdots, \cos(\boldsymbol{\omega}_M^\top \boldsymbol{x}) + \cos(\boldsymbol{\omega}_M'^\top \boldsymbol{x})]^\top$, and $\phi_{ns,c}(\boldsymbol{x}) = [\cos(\boldsymbol{\omega}_1^\top \boldsymbol{x}) + \cos(\boldsymbol{\omega}_1'^\top \boldsymbol{x}), \cdots, \cos(\boldsymbol{\omega}_M^\top \boldsymbol{x}) + \cos(\boldsymbol{\omega}_M'^\top \boldsymbol{x}), \sin(\boldsymbol{\omega}_1^\top \boldsymbol{x}) + \sin(\boldsymbol{\omega}_1'^\top \boldsymbol{x}), \cdots, \sin(\boldsymbol{\omega}_M^\top \boldsymbol{x}) + \sin(\boldsymbol{\omega}_M'^\top \boldsymbol{x})]^\top$, respectively, and $\{\boldsymbol{\omega}_i, \boldsymbol{\omega}_i'\}_{i=1}^M \overset{i.i.d.}{\sim} \mathcal{N}(\boldsymbol{0}, \sigma^2 \boldsymbol{I})$. The corresponding deep spectral kernels $k_{ns,w/o}^L(\boldsymbol{x}, \boldsymbol{x}')$ and $k_{ns,c}^L(\boldsymbol{x}, \boldsymbol{x}')$ is formulated as:*

$$k_{ns,w/o}^L(\boldsymbol{x}, \boldsymbol{x}') = C_{ns,w/o}^{L-1}(\cosh(\sigma^2(k_{ns,w/o}^{L-1}(\boldsymbol{x}, \boldsymbol{x}'))) + 2), L \geq 2,$$

$$C_{ns,w/o}^{L-1} = e^{-\frac{\|\phi_{ns,w/o}^{L-1}\cdots\phi_{ns,w/o}^1(\boldsymbol{x})\|_2^2 + \|\phi_{ns,w/o}^{L-1}\cdots\phi_{ns,w/o}^1(\boldsymbol{x}')\|_2^2}{2}\sigma^2}, \quad (8)$$

*and*

$$k_{ns,c}^L(\boldsymbol{x}, \boldsymbol{x}') = C_{ns,c}^{L-1}(\exp(\sigma^2(k_{ns,c}^{L-1}(\boldsymbol{x}, \boldsymbol{x}'))) + 1), L \geq 2,$$

$$C_{ns,c}^{L-1} = 2e^{-\frac{\|\phi_{ns,c}^{L-1}\cdots\phi_{ns,c}^1(\boldsymbol{x})\|_2^2 + \|\phi_{ns,c}^{L-1}\cdots\phi_{ns,c}^1(\boldsymbol{x}')\|_2^2}{2}\sigma^2}. \quad (9)$$

**Proof 2** *The proof is relegated to the appendix of our paper due to space limitations.*

Proposition 1 and Proposition 2 demonstrate that deep spectral kernels in the removal and concatenation scenarios are constructed through hierarchical stacking, governed by the functions $\cosh(\cdot)$ and $\exp(\cdot)$, respectively. These functions instantiate the role of $g(\cdot)$ in equation 5.

**Remark 1 (Answer for the first question)** *The hierarchical stacking of deep spectral kernels can be governed by the functions $\cosh(\cdot)$ for the removal and $\exp(\cdot)$ for the concatenation. Formally, both $\cosh(\cdot)$ and $\exp(\cdot)$ are holomorphic over $\mathbb{C}$, with nonnegative coefficients in their Maclaurin series. Consequently, by invoking Lemma 1, we conclude that **all four classes of deep spectral kernels retain the reproducing property throughout the stacking process.***

### 3.2 RKHS EXPANSION OF DEEP SPECTRAL KERNELS

A reproducing kernel $k(\cdot, \cdot)$ defined on a dataset $\boldsymbol{X} = \{\boldsymbol{x}_i\}_{i=1}^N$ induces a corresponding RKHS $\mathcal{H}_k$. The expansion of this RKHS can be formally described as an inclusion relation between the original and its expansion. The inclusion relation between two RKHSs, $\mathcal{H}_{k_1}$ and $\mathcal{H}_{k_2}$, induced by kernels $k_1(\cdot, \cdot)$ and $k_2(\cdot, \cdot)$ respectively, was first examined in Aronszajn (1950). This work established a connection between the inclusion relation of two RKHSs and the relationships between their associated kernels, as summarized in Lemma 2. Subsequently, Huang *et al.,* Huang et al. (2023) extended this result to hierarchical kernels, where $k_2$ is constructed as $g(k_1)$ with $g(\cdot)$ constrained to be holomorphic. Formally, denote $k_1 \preceq k_2$ if $k_2 - k_1$ remains a kernel.

**Lemma 2** *Let $k_1(\cdot, \cdot)$ and $k_2(\cdot, \cdot)$ be two kernels. Then $\mathcal{H}_{k_1} \subset \mathcal{H}_{k_2}$ if and only if there exists a non-negative constant $\alpha$ such that $k_1 \preceq \alpha k_2$.*

**Lemma 3** *Huang et al. (2023) Let $g$ be a holomorphic function on $\mathbb{C}$ of the form*

$$g(z) = \sum_{q=0}^{\infty} a_q z^q, z \in \mathbb{C}, a_q \geq 0, q \in \mathbb{Z}, \quad (10)$$

*where $a_1 > 0$. Then $\mathcal{H}_k \subset \mathcal{H}_{g(k)}$. In particular, $\mathcal{H}_k \subseteq \mathcal{H}_{\exp(k)}$.*

In line with the formulation provided in Section 2.2 and equation 5, Lemma 2 and Lemma 3 allow the inclusion relation between $\mathcal{H}_{k^L}$ and $\mathcal{H}_{k^{L-1}}$ can be transformed as the relationship between deep spectral kernels $k^L$ and $k^{L-1}$, which are characterized in Proposition 1 and Proposition 2. Since the coefficients $C_{s,w/o}^{L-1}$, $C_{s,c}^{L-1}$, $C_{ns,w/o}^{L-1}$, $C_{ns,c}^{L-1}$, and $\delta^2$ are strictly positive, they do not affect the intrinsic relationships between deep spectral kernels. Accordingly, the analysis reduces to verifying the conditions of Lemma 2 and Lemma 3 for $k^L = g(k^{L-1})$.

For the concatenation scenario, the relationship between $k_{s,c}^L$ and $k_{s,c}^{L-1}$, as well as that between $k_{ns,c}^L$ and $k_{ns,c}^{L-1}$ can be simplified as $k_{s,c}^L = \exp(k_{s,c}^{L-1})$ and $k_{ns,c}^L = \exp(k_{ns,c}^{L-1})$ for $L \geq 2$. By and Lemma 3, it follows directly that $\mathcal{H}_{k_{s,c}^{L-1}} \subseteq \mathcal{H}_{\exp(k_{s,c}^{L-1})} = \mathcal{H}_{k_{s,c}^L}$ and $\mathcal{H}_{k_{ns,c}^{L-1}} \subseteq \mathcal{H}_{\exp(k_{ns,c}^{L-1})} = \mathcal{H}_{k_{ns,c}^L}$, implying that the RKHS is expanding as the spectral kernel depth increases.

For the removal scenario, the relationships $k_{s,w/o}^L = \cosh(k_{s,w/o}^{L-1})$ and $k_{ns,w/o}^L = \cosh(k_{ns,w/o}^{L-1})$ hold for $L \geq 2$. Without loss of generality, we assume $k^L = \cosh(k^{L-1}), L \geq 2$ for a general deep spectral kernel, such that:

$$k^L = \cosh(k^{L-1}) = \sum_{p=0}^{\infty} \frac{(k^{L-1})^{2p}}{(2p)!} = 1 + \frac{(k^{L-1})^2}{2!} + \frac{(k^{L-1})^4}{4!} + \cdots. \tag{11}$$

Next, we investigate the existence of a non-negative constant $\alpha$ holding $k^{L-1} \preceq \alpha k^L$. If such an $\alpha$ exists, then the difference $\alpha k^L - k^{L-1} = \alpha\left(1 + \frac{(k^{L-1})^2}{2!} + \frac{(k^{L-1})^4}{4!} + \cdots\right) - k^{L-1}$ must remains a reproducing kernel. Because $1 + \frac{(k^{L-1})^2}{2!} + \frac{(k^{L-1})^4}{4!} + \cdots$ does not contain the linear term $k^{L-1}$, the negative term $-k^{L-1}$ inevitably retains in $\alpha k^L - k^{L-1}$ for any $\alpha$. This violates the positive definiteness of $\alpha k^L - k^{L-1}$, and thus no such $\alpha$ can satisfy $k^{L-1} \preceq \alpha k^L$. We therefore conclude that the RKHS fails to exhibit progressive expansion with increasing spectral kernel depth.

**Remark 2 (Answer for the second question)** *When the imaginary component is preserved with the concatenation form, the RKHS associated with deep spectral kernels exhibits progressive expansion with increasing spectral kernel depth. Conversely, this property ceases to hold upon removal of the imaginary component.*

In summary, this section has investigated the general characterizations of deep spectral kernels, including the reproducing property and the RKHS extension. Our theoretical analyses indicate that deep spectral kernels consistently retain the reproducing property during the stacking process. However, their associated RKHSs exhibit progressive expansion with increasing depth only when the imaginary component is preserved, revealing the essential role of imaginary components.

## 4 GENERATIVE SPECTRAL KERNEL

The RKHS associated with the deep spectral kernel under the concatenation scenario exhibits progressive expansion, thereby endowing the model with enhanced representational capacity with the increasing depth. Nevertheless, these methods still suffer from limitations: (1) the optimization challenge. By stacking the periodic functions (*i.e.,* the spectral kernel mappings), the deep spectral kernel is prone to local minima. (2) The range of spectral density. In most existing deep spectral kernel models, the spectral density function is data-independent and consists of predetermined components, restricting the range of spectral density even within deep architectures. For example, Huang *et al.* Huang et al. (2023) explored the deep Gaussian kernel and derived its associated spectral density function (detailed result is shown in Appendix B). The associated spectral density consists of predetermined Gaussian components specified by the chosen bandwidth and the number of layers.

In the spectral kernel perspective, the progressive expansion (inclusion relation) of RKHSs induced by the deep spectral kernels is linked to their corresponding spectral densities Huang et al. (2023); Zhang & Zhao (2011). Moreover, Theorems 1 and 2 establish a one-to-one correspondence between kernels and their spectral density through Fourier duality. These results indicate that a deep spectral kernel can be implemented by directly resampling from an adaptive spectral density. This motivates us to propose a novel deep spectral kernel learning framework, called **gen**erative **s**pectral **ker**nel (**GensKer**), to address these limitations. This method directly generates an adaptive spectral density and then constructs an associated spectral kernel based on Theorem 2. Concretely, GensKer includes two core modules: the spectral generation module and the spectral kernel construction module. The former is used to generate a joint spectral distribution $s(\boldsymbol{\omega}, \boldsymbol{\omega}')$. The latter constructs the spectral kernel using the sampled frequency pairs $\{\boldsymbol{\omega}_i, \boldsymbol{\omega}_i'\}_{i=1}^M \sim s(\boldsymbol{\omega}, \boldsymbol{\omega}')$ with the concatenation form. The overall architecture is shown in Figure 1. This method, with a single spectral kernel architecture, is capable of generating an adaptive and data-dependent spectral density, thereby realizing the deep spectral kernel construction. Furthermore, this architecture enables GensKer to circumvent the optimization challenges posed by deep spectral kernels.

Figure 1: The architecture of the generative spectral kernel. The spectral generation module is used to generate the spectral distribution $s(\boldsymbol{\omega}, \boldsymbol{\omega}')$. The loss $\mathcal{L}_p$ is introduced to constrain $s(\boldsymbol{\omega}'|\boldsymbol{\omega})s(\boldsymbol{\omega}) = s(\boldsymbol{\omega}|\boldsymbol{\omega}')s(\boldsymbol{\omega}')$. The spectral kernel is constructed via the generated frequency pairs $\{\boldsymbol{\omega}_i, \boldsymbol{\omega}'_i\}_{i=1}^M$.

## 4.1 SPECTRAL GENERATIVE MODULE

This module generates the spectral density using a spectral generation network. We redefine the spectral density $s(\boldsymbol{\omega}, \boldsymbol{\omega}')$ through the desired conditional distribution, i.e., $s(\boldsymbol{\omega}, \boldsymbol{\omega}') = s(\boldsymbol{\omega}'|\boldsymbol{\omega})s(\boldsymbol{\omega})$. Accordingly, the non-stationary spectral kernel $k(\boldsymbol{x}, \boldsymbol{x}')$ in equation 2 can be reformulated by:

$$k(\boldsymbol{x}, \boldsymbol{x}') = \int_{\mathbb{R}^d} s(\boldsymbol{\omega}'|\boldsymbol{\omega})s(\boldsymbol{\omega})e^{i(\boldsymbol{\omega}^\top \boldsymbol{x} - \boldsymbol{\omega}'^\top \boldsymbol{x}')}d\boldsymbol{\omega}d\boldsymbol{\omega}'. \tag{12}$$

Here $s(\boldsymbol{\omega}'|\boldsymbol{\omega})s(\boldsymbol{\omega})$ is obtained via a spectral generation network, which is formulated as follows:

$$s(\boldsymbol{\omega}'|\boldsymbol{\omega})s(\boldsymbol{\omega}) = s(\boldsymbol{\omega}'|\boldsymbol{\omega} = G_2(\boldsymbol{\epsilon}_2))s(\boldsymbol{\omega}|\boldsymbol{\epsilon}_2)p(\boldsymbol{\epsilon}_2),$$

$$\boldsymbol{\omega} = G_2(\boldsymbol{\epsilon}_2) \sim s(\boldsymbol{\omega}|\boldsymbol{\epsilon}_2)p(\boldsymbol{\epsilon}_2), \quad (\boldsymbol{\omega}'|\boldsymbol{\omega} = G_2(\boldsymbol{\epsilon}_2)) = G_2(\boldsymbol{\epsilon}_2) \odot B_1(\boldsymbol{\epsilon}_1) + \Upsilon_1(\boldsymbol{\epsilon}_1), \tag{13}$$

where $\boldsymbol{\epsilon}_1, \boldsymbol{\epsilon}_2 \overset{\text{i.i.d.}}{\sim} p(\boldsymbol{\epsilon})$ are the initialized spectrum and are sampled from the initialized spectral density $p(\boldsymbol{\epsilon})$. $G_2$ denotes a generator, making $\boldsymbol{\omega} = G_2(\boldsymbol{\epsilon}_2) \sim s(\boldsymbol{\omega})$. $B_1$ and $\Upsilon_1$ represent encoders, using to learn the conditional representation $(\boldsymbol{\omega}'|\boldsymbol{\omega} = G_2(\boldsymbol{\epsilon}_2))$.

Since the joint spectral density $s(\boldsymbol{\omega}, \boldsymbol{\omega}')$ can be equivalently expressed via both $s(\boldsymbol{\omega}'|\boldsymbol{\omega})s(\boldsymbol{\omega})$ and $s(\boldsymbol{\omega}|\boldsymbol{\omega}')s(\boldsymbol{\omega}')$, the non-stationary spectral kernel can also be redefined as:

$$k(\boldsymbol{x}, \boldsymbol{x}') = \int_{\mathbb{R}^d} s(\boldsymbol{\omega}|\boldsymbol{\omega}')s(\boldsymbol{\omega}')e^{i(\boldsymbol{\omega}^\top \boldsymbol{x} - \boldsymbol{\omega}'^\top \boldsymbol{x}')}d\boldsymbol{\omega}d\boldsymbol{\omega}',$$

$$s(\boldsymbol{\omega}|\boldsymbol{\omega}')s(\boldsymbol{\omega}') = s(\boldsymbol{\omega}|\boldsymbol{\omega}' = G_1(\boldsymbol{\epsilon}_1))s(\boldsymbol{\omega}'|\boldsymbol{\epsilon}_1)p(\boldsymbol{\epsilon}_1), \tag{14}$$

$$\boldsymbol{\omega}' = G_1(\boldsymbol{\epsilon}_1) \sim s(\boldsymbol{\omega}|\boldsymbol{\epsilon}_1)p(\boldsymbol{\epsilon}_1), \quad (\boldsymbol{\omega}|\boldsymbol{\omega}' = G_1(\boldsymbol{\epsilon}_1)) = G_1(\boldsymbol{\epsilon}_1) \odot B_2(\boldsymbol{\epsilon}_2) + \Upsilon_2(\boldsymbol{\epsilon}_2),$$

where $G_1$ represents a generator, making $\boldsymbol{\omega}' = G_1(\boldsymbol{\epsilon}_1) \sim s(\boldsymbol{\omega}')$. $B_2$ and $\Upsilon_2$ are encoders to learn the conditional representation $(\boldsymbol{\omega}|\boldsymbol{\omega}' = G_1(\boldsymbol{\epsilon}_1))$.

To ensure $s(\boldsymbol{\omega}'|\boldsymbol{\omega})s(\boldsymbol{\omega}) = s(\boldsymbol{\omega}|\boldsymbol{\omega}')s(\boldsymbol{\omega}')$, we enforce a loss $\mathcal{L}_p$. Specifically, this loss enables $s(\boldsymbol{\omega}'|\boldsymbol{\omega} = G_2(\boldsymbol{\epsilon}_2))s(\boldsymbol{\omega}|\boldsymbol{\epsilon}_2)p(\boldsymbol{\epsilon}_2) = s(\boldsymbol{\omega}|\boldsymbol{\omega}' = G_1(\boldsymbol{\epsilon}_1))s(\boldsymbol{\omega}'|\boldsymbol{\epsilon}_1)p(\boldsymbol{\epsilon}_1)$.

The generated $s(\boldsymbol{\omega}, \boldsymbol{\omega}')$ is a probability density function, satisfying the positive definiteness condition of the spectral density in Theorem 2. Therefore, it ensures that the constructed spectral kernels are positive definite, thereby endowing them with the reproducing property.

## 4.2 SPECTRAL KERNEL MODULE

Building upon the theoretical analysis in Section 3.2, this module applies the generated spectral density to construct spectral kernels under the concatenation form, coupling it with the spectral gen-

eration module to form a unified model. This end-to-end framework enables the proposed GensKer to generate a data-dependent and adaptive spectral density, resulting in an adaptive RKHS.

As presented in Section 2.2, the non-stationary spectral kernel can be defined by:

$$k(\boldsymbol{x}, \boldsymbol{x}') \approx \langle \phi_{ns,c}(\boldsymbol{x}), \phi_{ns,c}(\boldsymbol{x}') \rangle, \quad \phi_{ns,c}(\boldsymbol{x}) = \frac{1}{\sqrt{4M}} \begin{bmatrix} \cos(\boldsymbol{\Omega}^\top \boldsymbol{x}) + \cos(\boldsymbol{\Omega}'^\top \boldsymbol{x}) \\ \sin(\boldsymbol{\Omega}^\top \boldsymbol{x}) + \sin(\boldsymbol{\Omega}'^\top \boldsymbol{x}) \end{bmatrix}, \quad (15)$$

where $\boldsymbol{\Omega} = [\boldsymbol{\omega}_1, \boldsymbol{\omega}_2, \cdots, \boldsymbol{\omega}_M]$ and $\boldsymbol{\Omega}' = [\boldsymbol{\omega}'_1, \boldsymbol{\omega}'_2, \cdots, \boldsymbol{\omega}'_M]$ are frequency matrices. Instead of learning the frequency pairs through hierarchically stacked spectral kernel mappings, we sample them directly from the generated spectral density. In particular, $\{\boldsymbol{\omega}_i\}_{i=1}^M$ are sampled from the conditional distribution $s(\boldsymbol{\omega}|\boldsymbol{\omega}')$, and $\{\boldsymbol{\omega}'_i\}_{i=1}^M$ are sampled from $s(\boldsymbol{\omega}'|\boldsymbol{\omega})$. $M$ is the sampling number.

As a result, we obtain a generative spectral kernel, which is constructed by the generated spectral density. Taking the classification task as an example, where the spectral kernel mapping $\phi_{ns,c}(\boldsymbol{x})$ is considered as the feature extractor. The loss function is defined as: $\mathcal{L} = \mathcal{L}_c + \mathcal{L}_p$, where $\mathcal{L}_p = \ell_p\Big(G_2(\boldsymbol{\epsilon}_2) \odot (G_2(\boldsymbol{\epsilon}_2) \odot B_1(\boldsymbol{\epsilon}_1) + \Upsilon_1(\boldsymbol{\epsilon}_1)), G_1(\boldsymbol{\epsilon}_1)(G_1(\boldsymbol{\epsilon}_1) \odot B_2(\boldsymbol{\epsilon}_2) + \Upsilon_2(\boldsymbol{\epsilon}_2))\Big)$ denotes the distribution loss to constraint $s(\boldsymbol{\omega}'|\boldsymbol{\omega} = G_2(\boldsymbol{\epsilon}_2))s(\boldsymbol{\omega}|\boldsymbol{\epsilon}_2)p(\boldsymbol{\epsilon}_2) = s(\boldsymbol{\omega}|\boldsymbol{\omega}' = G_1(\boldsymbol{\epsilon}_1))s(\boldsymbol{\omega}'|\boldsymbol{\epsilon}_1)p(\boldsymbol{\epsilon}_1)$. $\mathcal{L}_c = \frac{1}{N}\sum_{i=1}^N \ell_c(\boldsymbol{x}_i, y_i)$ denotes the classification loss.

## 5 EXPERIMENTS

This section experimentally validates the theoretical results (*i.e.,* the influence of the imaginary component on the RKHS and performance) and the principal advantages of the proposed GensKer. All experiments are implemented using PyTorch Paszke et al. (2019) and conducted on a workstation with NVIDIA RTX 3090 GPU, AMD R7-5700X 3.40GHz 8-core CPU, and 32 GB memory.

### 5.1 EXPERIMENTAL VALIDATION FOR THE THEORETICAL RESULTS IN SECTION 3.2

#### 5.1.1 THE INFLUENCE OF THE IMAGINARY COMPONENT ON THE RKHS

The theoretical analysis in Section 3.2 demonstrates that the RKHS associated with deep spectral kernels, constructed with the concatenation form, exhibits progressive expansion with increasing depth. Conversely, this property ceases to hold upon removal of the imaginary components. In this part, we verify this point by examining the relationship between the performance of deep spectral kernels and their depth in kernel regression tasks, which relatively reflects the theoretical properties of

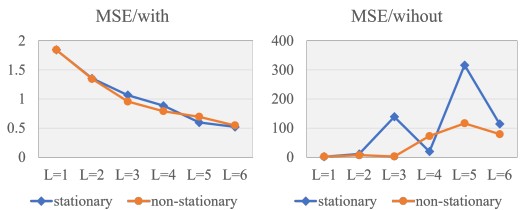

Figure 2: The performance (MSE) changes.

these kernels. We first elaborate a series of synthetic data $\{x_i, y_i\}_{i=1}^{6000}$ by the function $y_i = \sin(x_i) + \sin(3x_i) + \sin(5x_i) + \sin(10x_i)$, $x_i \in [0, 10]$, where 5000 samples are used for training and 1000 for testing. Next, we investigate the performance changes of deep spectral kernel models with increasing depth on the synthetic data. Considering the optimization challenges and the risk of overfitting, the maximum number of $L$ is set to 6. The result is shown in Figure 2.

The result in Figure 2 illustrates that the performance under the concatenation scenario (*i.e.,* with the imaginary component) achieves incremental improvement with increasing depth for both stationary and non-stationary cases. This is attributed to the integration of the imaginary component, resulting in a progressive extension of the RKHS. By contrast, the performance under the removal scenario (*i.e.,* without the imaginary component) exhibits the instability for both cases. This is evidenced by the fact that the performance appears to have no statistically significant correlation with the model depth. These findings align with our theoretical analysis.

#### 5.1.2 THE INFLUENCE OF THE IMAGINARY COMPONENT ON THE PERFORMANCE

In fact, the spectral kernel mapping can be interpreted as a single-layer neural network with the $\sin(\cdot)$ or $\cos(\cdot)$ activation. By stacking such spectral kernel mappings, one obtains $\phi^L(\phi^{L-1}\cdots\phi^1(\cdot))$,

which constitutes a deep spectral kernel network. In this experiment, we investigate the impact of the imaginary component by comparing two forms (with or without the imaginary component) of deep spectral kernel networks on the time series classification task based on 6-sub-datasets with default training and testing data splitting from the **UCR Archive** Dau et al. (2019) dataset. The deep spectral kernel network with $L = 3$ is considered as the feature extractor, and a fully connected layer serves as the classifier. All models are trained using ADAM Kingma & Ba (2014) with a learning rate of 0.01. Results are presented in Table 1.

The quantitative results reveal that: (1) The concatenation scenario (*i.e.,* with the imaginary component) consistently shows outstanding performance on both stationary and non-stationary cases. This is attributed to the integration of the imaginary component, which can mine more complex patterns of data. (2) The deep spectral kernel, constructed from the non-stationary spectral kernel, outperforms the one induced by a stationary spectral kernel on most

Table 1: Time series classification results under both stationary (S) and non-stationary (Ns) cases. w/ denotes the case of combining the imaginary component in the concatenation way, while w/o denotes the case of removing the imaginary component. ↑ represents the performance improvement brought by the imaginary component.

| | | FordA | ECG200 | ECG5000 | Rock | Trace | Plane |
|---|---|---|---|---|---|---|---|
| | w/ | 83.42 | 92.40 | 93.63 | 66.60 | 83.00 | 97.81 |
| S | w/o | 82.59 | 89.50 | 93.60 | 65.50 | 82.53 | 96.76 |
| | ↑ | 0.83 | 2.90 | 0.03 | 1.10 | 0.47 | 1.05 |
| | w/ | 83.99 | 92.20 | 93.54 | 68.20 | 83.70 | 98.00 |
| Ns | w/o | 83.66 | 91.80 | 93.31 | 63.50 | 82.90 | 97.19 |
| | ↑ | 0.33 | 0.40 | 0.23 | 4.70 | 0.80 | 0.81 |

datasets. This confirms that non-stationary spectral kernels are capable of breaking the locality limitations in the stationary case while effectively capturing long-range dependencies within the data.

## 5.2 EXPERIMENTAL VALIDATION FOR THE PRINCIPAL ADVANTAGES OF GENSKER

### 5.2.1 VALIDATION FOR THE ADVANTAGE IN ADAPTIVE SPECTRAL DENSITY

As advocated in Section 4, GensKer with a single-layer spectral kernel architecture is capable of generating an adaptive spectral density and realizing the deep spectral kernel. In this section, time series classification tasks on the **UCR Archive** dataset are also performed to verify these points. On the one hand, we compare the plain deep spectral kernel models (under the concatenation scenario without any additional conditions) with their corresponding generative spectral kernel models for both stationary (S) and non-stationary (Ns) cases. On the other hand, we compare our method with several state-of-the-art spectral kernel learning methods, including **TRF** Shilton et al. (2022), **GRFF** Fang et al. (2023), and **DKEF** Wenliang et al. (2019) for the stationary case, as well as **DSKN** Xue et al. (2019), **ASKL** Li et al. (2020), and **CokeNet** Tian et al. (2024) for the non-stationary case. The detailed information of these methods can be found in Appendix C. For the plain deep spectral kernel models, a three-layer deep spectral kernel network is used for inherent feature extraction. For the generative spectral kernel models, to avoid the optimization challenges posed by the deep spectral kernel models, the generators ($G_1$, $G_2$) and encoders ($B_1$, $B_2$, $\Upsilon_1$, $\Upsilon_2$) are implemented as simple fully connected networks with ReLU activation. Specifically, $G_1$, $G_2$, $B_1$, and $B_2$ include one hidden layer, while $\Upsilon_1$ and $\Upsilon_2$ are the linear transformation. A three-layer fully connected network with ReLU activation serves as the classifier for both types of methods. All models are trained by using ADAM with a learning rate of 0.01 (or 0.001). The results are reported in Table 2.

Table 2: Time series classification results (accuracy (Acc) and parameters (Para)) for the stationary (S) and non-stationary (Ns) cases. The best results are highlighted in **bold**.

| | Model | FordA | | ECG200 | | ECG5000 | | Rock | | Trace | | Plane | |
|---|---|---|---|---|---|---|---|---|---|---|---|---|---|
| | | Acc | Para | Acc | Para | Acc | Para | Acc | Para | Acc | Para | Acc | Para |
| | plain | 74.28 | 1.59M | 90.70 | 1.38M | 93.05 | 1.40M | 64.80 | 2.79M | 81.70 | 1.47M | 95.81 | 1.41M |
| | TRF | 65.74 | 0.52M | 88.60 | 0.32M | 91.72 | 0.34M | 59.60 | 1.72M | 74.90 | 0.41M | 94.57 | 0.34M |
| S | GRFF | 74.92 | **0.28M** | 85.90 | **0.23M** | 93.10 | **0.24M** | 69.60 | **0.59M** | 81.70 | **0.25M** | 96.76 | **0.24M** |
| | DKEF | 76.65 | 0.65M | 87.80 | 0.37M | 91.94 | 0.40M | 59.20 | 1.81M | 75.70 | 0.47M | 96.38 | 0.39M |
| | Ours | **79.42** | 0.43M | **91.70** | 0.32M | **93.14** | 0.33M | **71.40** | 1.03M | **82.90** | 0.37M | **96.89** | 0.34M |
| | plain | 76.11 | 2.89M | **92.00** | 2.48M | 93.09 | 2.53M | 65.20 | 5.29M | 82.10 | 2.66M | 95.71 | 2.53M |
| | DSKN | 75.47 | 1.72M | 90.00 | 1.30M | 92.78 | 1.35M | 59.60 | 4.12M | 81.70 | 1.49M | 94.95 | 1.35M |
| Ns | ASKL | 72.50 | 2.00M | 87.53 | **0.39M** | 92.75 | 0.57M | 44.00 | 5.70M | 80.00 | 1.11M | 97.14 | 0.59M |
| | CokeNet | 71.61 | 1.58M | 89.95 | 0.76M | 93.40 | 0.85M | 57.50 | 6.39M | 76.05 | 1.12M | 97.14 | 0.86M |
| | Ours | **82.04** | **0.79M** | 91.70 | 0.42M | **93.59** | **0.46M** | **72.60** | 3.31M | **83.60** | **0.58M** | **97.43** | **0.46M** |

The results in Table 2 illustrate the following information: (1) the generative spectral kernel model achieves performance comparable to, and even superior to, the plain deep spectral kernel under both stationary and non-stationary cases. This indicates that our proposal is capable of realizing the deep spectral kernel by a single-layer spectral kernel architecture. (2) Compared with other SOTA

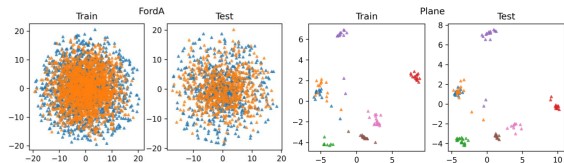

Figure 3: The data distributions of **FordA** and **Plane**.

spectral kernel learning methods, our proposed generative spectral kernel also achieves outstanding performance with fewer parameters, especially for the non-stationary case. This reveals that the proposed GensKer is capable of generating an adaptive spectral density and further exploring the inherent pattern within the data. Specifically, our method performs exceptionally well on **FordA**, while its performance on **Plane** is comparable to other models. We visualize their data distribution in Figure 3 and can observe that the classes in **Plane** are well-separated and exhibit high intra-class compactness, resulting in similar performance for all methods. By contrast, there is no significant difference between the classes in **FordA**. Therefore, compelling further analysis is needed to elucidate the divergence in their latent patterns, highlighting the effectiveness of our method in generating a data-dependent spectral density and uncovering the essential characteristics of the data. The data distributions of the remaining datasets and analyses are shown in Appendix D.1. In summary, our method effectively captures the inherent data patterns through the adaptive spectral density while requiring fewer parameters, thereby yielding a more lightweight model with broader applicability.

### 5.2.2 VALIDATION FOR THE ADVANTAGE IN OF

In this experiment, we evaluate the ability of GensKer in circumventing the optimization challenges by comparing the geometry of the loss landscape and the convergence trained by ADAM on **FordA**. We visualize the loss landscape around the final epoch and the training loss curve for the deep spectral kernel and generative spectral kernel in Figure 4. It demonstrates that (1) the loss landscape of the deep spectral kernel is rugged with a sharp minimum. By contrast, the proposed GensKer exhibits a smooth loss landscape with a flat minimum. (2) The training loss of the deep spectral kernel exhibits oscillations for the first 15 epochs and then rapidly converges to a sharp minimum. In comparison, our method descends smoothly without instability and converges to a lower loss value. These results are

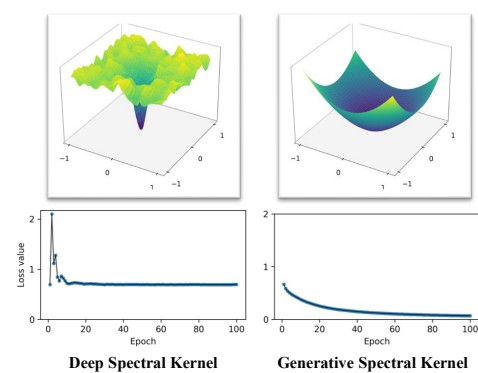

Figure 4: The loss landscape and curve.

consistent with their loss landscape. All the observations demonstrate that the proposed GensKer effectively mitigates the optimization challenge posed by deep spectral kernels.

## 6 CONCLUSION

In this paper, we first investigate two questions concerning the deep spectral kernel: (1) Does the deep spectral kernel retain the reproducing property during the stacking process? (2) In which class can the RKHS induced by the deep spectral kernel expand with increasing depth? Rigorous theoretical analyses indicate that all types of deep spectral kernels maintain the reproducing property throughout the stacking process. Furthermore, when retaining the imaginary component, the RKHS corresponding to the deep spectral kernel expands with increasing spectral kernel depth, which is related to the spectral density function from the perspective of the spectral kernel. Based on these findings, we propose a generative spectral kernel, which directly generates the spectral density via a generative network. This method, with a single-layer spectral kernel architecture, can: (1) generate an adaptive sepctrl density and achieve deep spectral kernel performance; (2) circumvent the optimization challenges introduced by stacked periodic functions. Finally, these findings have been verified through a set of experiments using synthetic data and six real-world time series datasets.

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

In this section, we provide:

- Detailed proof of theoretical results, including Proposition 1, Proposition 2.

- The mentioned special case, deep Gaussian kernel, in Section 4 of the main paper.

- The detailed information of the compared SOTA spectral kernel learning methods.

- The additional experiment results. The data distribution, loss landscape for the other datasets, and further evaluation of the generated spectral density.

- The statement of the use of large language models (LLMs).

## A  DETAILED PROOF OF THEORETICAL RESULTS

### A.1  PROOF OF PROPOSITION 1

**Proposition 1** *For the stationary spectral kernel under the removal and concatenation scenarios, the spectral kernel mappings are defined by:* $\phi_{s,w/o}(\boldsymbol{x}) = [\cos(\boldsymbol{\omega}_1^\top \boldsymbol{x}), \cdots, \cos(\boldsymbol{\omega}_M^\top \boldsymbol{x})]^\top$ *and* $\phi_{s,c}(\boldsymbol{x}) = [\cos(\boldsymbol{\omega}_1^\top \boldsymbol{x}), \cdots, \cos(\boldsymbol{\omega}_M^\top \boldsymbol{x}), \sin(\boldsymbol{\omega}_1^\top \boldsymbol{x}), \cdots, \sin(\boldsymbol{\omega}_M^\top \boldsymbol{x})]^\top$, *respectively, and* $\{\boldsymbol{\omega}_i\}_{i=1}^M \sim \mathcal{N}(\boldsymbol{0}, \sigma^2 \boldsymbol{I})$. *The stacking of their corresponding deep spectral kernels* $k_{s,w/o}^L(\boldsymbol{x}, \boldsymbol{x}')$ *and* $k_{s,c}^L(\boldsymbol{x}, \boldsymbol{x}')$ *is formulated as:*

$$k_{s,w/o}^L(\boldsymbol{x}, \boldsymbol{x}') = C_{s,w/o}^{L-1} \cosh(\sigma^2(k_{s,w/o}^{L-1}(\boldsymbol{x}, \boldsymbol{x}'))), L \geq 2,$$
$$C_{s,w/o}^{L-1} = \frac{1}{4}[e^{-\frac{\|\phi_{s,w/o}^{L-1}\cdots\phi_{s,w/o}^1(\boldsymbol{x})\|_2^2 + \|\phi_{s,w/o}^{L-1}\cdots\phi_{s,w/o}^1(\boldsymbol{x}')\|_2^2}{2}\sigma^2}], \tag{1}$$

*and*

$$k_{s,c}^L(\boldsymbol{x}, \boldsymbol{x}') = C_{s,c}^{L-1} \exp(\sigma^2(k_{s,c}^{L-1}(\boldsymbol{x}, \boldsymbol{x}'))), L \geq 2,$$
$$C_{s,c}^{L-1} = \frac{1}{2}[e^{-\frac{\|\phi_{s,c}^{L-1}\cdots\phi_{s,c}^1(\boldsymbol{x})\|_2^2 + \|\phi_{s,c}^{L-1}\cdots\phi_{s,c}^1(\boldsymbol{x}')\|_2^2}{2}\sigma^2}]. \tag{2}$$

**Proof 1** *Let* $k_{s,w/o}^L(\boldsymbol{x}, \boldsymbol{x}') = \langle\phi_{s,w/o}^L(\cdots\phi_{s,w/o}^1(\boldsymbol{x})), \phi_{s,w/o}^L(\cdots\phi_{s,w/o}^1(\boldsymbol{x}'))\rangle$ *denotes the deep spectral kernel with L layers under the removal scenario, where* $\phi_{s,w/o}^1(\boldsymbol{x}) = [\cos(\boldsymbol{\omega}_1^{1\top}\boldsymbol{x}), \cdots, \cos(\boldsymbol{\omega}_M^{1\top}\boldsymbol{x})]^\top$ *and* $\phi_{s,w/o}^1(\boldsymbol{x}') = [\cos(\boldsymbol{\omega}_1^{1\top}\boldsymbol{x}'), \cdots, \cos(\boldsymbol{\omega}_M^{1\top}\boldsymbol{x}')]^\top$. *The hierarchical relationship of spectral kernel mapping is formulated by* $\phi_{s,w/o}^L(\boldsymbol{x}) = [\cos(\boldsymbol{\omega}_1^{L\top}(\phi_{s,w/o}^{L-1}(\boldsymbol{x}))), \cdots, \cos(\boldsymbol{\omega}_M^{L\top}(\phi_{s,w/o}^{L-1}(\boldsymbol{x})))]^\top$. *Following these settings, we have*

$$
\begin{aligned}
k_{s,w/o}^1(\boldsymbol{x}, \boldsymbol{x}') &= \langle\phi_{s,w/o}^1(\boldsymbol{x}), \phi_{s,w/o}^1(\boldsymbol{x}')\rangle \\
&= \frac{1}{2M}\sum_{i=1}^M \cos(\boldsymbol{\omega}_i^{1\top}\boldsymbol{x})\cos(\boldsymbol{\omega}_i^{1\top}\boldsymbol{x}') \\
&\approx \frac{1}{2}\mathbb{E}_{\boldsymbol{\omega}^1 \sim \mathcal{N}(\boldsymbol{0},\sigma^2\boldsymbol{I})}[\cos(\boldsymbol{\omega}^{1\top}\boldsymbol{x})\cos(\boldsymbol{\omega}^{1\top}\boldsymbol{x}')] \\
&= \frac{1}{2}\mathbb{E}_{\boldsymbol{\omega}^1 \sim \mathcal{N}(\boldsymbol{0},\sigma^2\boldsymbol{I})}\frac{1}{2}[\cos(\boldsymbol{\omega}^{1\top}\boldsymbol{x} + \boldsymbol{\omega}^{1\top}\boldsymbol{x}') + \cos(\boldsymbol{\omega}^{1\top}\boldsymbol{x} - \boldsymbol{\omega}^{1\top}\boldsymbol{x}')] \\
&= \frac{1}{4}[e^{-\frac{\|\boldsymbol{x}+\boldsymbol{x}'\|_2^2\sigma^2}{2}} + e^{-\frac{\|\boldsymbol{x}-\boldsymbol{x}'\|_2^2\sigma^2}{2}}] \\
&= \frac{1}{4}[e^{-\frac{[\|\boldsymbol{x}\|_2^2+\|\boldsymbol{x}'\|_2^2]\sigma^2}{2}}e^{-\sigma^2\boldsymbol{x}^\top\boldsymbol{x}'} + e^{-\frac{[\|\boldsymbol{x}\|_2^2+\|\boldsymbol{x}'\|_2^2]\sigma^2}{2}}e^{\sigma^2\boldsymbol{x}^\top\boldsymbol{x}'}] \\
&= \frac{1}{4}[e^{-\frac{[\|\boldsymbol{x}\|_2^2+\|\boldsymbol{x}'\|_2^2]\sigma^2}{2}}][e^{-\sigma^2\boldsymbol{x}^\top\boldsymbol{x}'} + e^{\sigma^2\boldsymbol{x}^\top\boldsymbol{x}'}] \\
&= \frac{1}{4}[e^{-\frac{[\|\boldsymbol{x}\|_2^2+\|\boldsymbol{x}'\|_2^2]\sigma^2}{2}}]\cosh(\sigma^2\boldsymbol{x}^\top\boldsymbol{x}'),
\end{aligned}
\tag{3}
$$

*Then,*

$$k_{s,w/o}^2(\boldsymbol{x}, \boldsymbol{x}') = \langle \phi_{s,w/o}^2(\phi_{s,w/o}^1(\boldsymbol{x})), \phi_{s,w/o}^2(\phi_{s,w/o}^1(\boldsymbol{x}')) \rangle$$

$$= \frac{1}{2M} \sum_{i=1}^M \cos(\boldsymbol{\omega}_i^{2\top} \phi_{s,w/o}^1(\boldsymbol{x})) \cos(\boldsymbol{\omega}_i^{2\top} \phi_{s,w/o}^1(\boldsymbol{x}'))$$

$$\approx \frac{1}{2} \mathbb{E}_{\boldsymbol{\omega}^2 \sim \mathcal{N}(\boldsymbol{0}, \sigma^2 \boldsymbol{I})} [\cos(\boldsymbol{\omega}^{2\top} \phi_{s,w/o}^1(\boldsymbol{x})) \cos(\boldsymbol{\omega}^{2\top} \phi_{s,w/o}^1(\boldsymbol{x}'))] \qquad (4)$$

$$= \frac{1}{4} [e^{-\frac{[\|\phi_{s,w/o}^1(\boldsymbol{x})\|_2^2 + \|\phi_{s,w/o}^1(\boldsymbol{x}')\|_2^2]\sigma^2}{2}}] \cosh(\sigma^2 \phi_{s,w/o}^1(\boldsymbol{x})^\top \phi_{s,w/o}^1(\boldsymbol{x}')),$$

*where $\phi_{s,w/o}^1(\boldsymbol{x})^\top \phi_{s,w/o}^1(\boldsymbol{x}') = k_{s,w/o}^1(\boldsymbol{x}, \boldsymbol{x}')$.*

*By employing the recursive iteration method, we can obtain:*

$$k_{s,w/o}^L(\boldsymbol{x}, \boldsymbol{x}') = C_{s,w/o}^L \cosh(\delta^2 k_{s,w/o}^{L-1}(\boldsymbol{x}, \boldsymbol{x}')), L \geq 2,$$

$$C_{s,w/o}^{L-1} = \frac{1}{4} [e^{-\frac{\|\phi_{s,w/o}^{L-1} \cdots \phi_{s,w/o}^1(\boldsymbol{x})\|_2^2 + \|\phi_{s,w/o}^{L-1} \cdots \phi_{s,w/o}^1(\boldsymbol{x}')\|_2^2}{2}\sigma^2}]. \qquad (5)$$

*Similar to the removal scenario, let $k_{s,c}^L(\boldsymbol{x}, \boldsymbol{x}') = \langle \phi_{s,c}^L(\cdots \phi_{s,c}^1(\boldsymbol{x})), \phi_{s,c}^L(\cdots \phi_{s,c}^1(\boldsymbol{x}')) \rangle$ denotes the deep spectral kernel with $L$ layers under the concatenation scenario, where $\phi_{s,c}^1(\boldsymbol{x}) = [\cos(\boldsymbol{\omega}_1^{1\top} \boldsymbol{x}), \cdots, \cos(\boldsymbol{\omega}_M^{1\top} \boldsymbol{x}), \sin(\boldsymbol{\omega}_1^{1\top} \boldsymbol{x}), \cdots, \sin(\boldsymbol{\omega}_M^{1\top} \boldsymbol{x})]^\top$ and $\phi_{s,c}^1(\boldsymbol{x}') = [\cos(\boldsymbol{\omega}_1^{1\top} \boldsymbol{x}'), \cdots, \cos(\boldsymbol{\omega}_M^{1\top} \boldsymbol{x}'), \sin(\boldsymbol{\omega}_1^{1\top} \boldsymbol{x}'), \cdots, \sin(\boldsymbol{\omega}_M^{1\top} \boldsymbol{x}')]^\top$. The hierarchical relationship of spectral kernel mapping is formulated by $\phi_{s,c}^L(\boldsymbol{x}) = [\cos(\boldsymbol{\omega}_1^{L\top}(\phi_{s,c}^{L-1}(\boldsymbol{x}))), \cdots, \cos(\boldsymbol{\omega}_M^{L\top}(\phi_{s,c}^{L-1}(\boldsymbol{x})))]^\top, \sin(\boldsymbol{\omega}_1^{L\top}(\phi_{s,c}^{L-1}(\boldsymbol{x}))), \cdots, \sin(\boldsymbol{\omega}_M^{L\top}(\phi_{s,c}^{L-1}(\boldsymbol{x})))]^\top$. Following these settings, we have*

$$k_{s,c}^1(\boldsymbol{x}, \boldsymbol{x}') = \langle \phi_{s,c}^1(\boldsymbol{x}), \phi_{s,c}^1(\boldsymbol{x}') \rangle$$

$$= \frac{1}{2M} \sum_{i=1}^M [\cos(\boldsymbol{\omega}_i^{1\top} \boldsymbol{x}) \cos(\boldsymbol{\omega}_i^{1\top} \boldsymbol{x}') + \sin(\boldsymbol{\omega}_i^{1\top} \boldsymbol{x}) \sin(\boldsymbol{\omega}_i^{1\top} \boldsymbol{x}')]$$

$$= \frac{1}{2M} \sum_{i=1}^M \cos(\boldsymbol{\omega}_i^{1\top} (\boldsymbol{x} - \boldsymbol{x}')) \qquad (6)$$

$$\approx \frac{1}{2} \mathbb{E}_{\boldsymbol{\omega}^1 \sim \mathcal{N}(\boldsymbol{0}, \sigma^2 \boldsymbol{I})} [\cos(\boldsymbol{\omega}^{1\top} (\boldsymbol{x} - \boldsymbol{x}'))]$$

$$= \frac{1}{2} e^{\frac{\|\boldsymbol{x} - \boldsymbol{x}'\|_2^2 \delta^2}{2}}.$$

*Then, the deep stationary spectral kernel with $L = 2$ can be formulated as follows:*

$$k_{s,c}^2(\boldsymbol{x}, \boldsymbol{x}') = \langle \phi_{s,c}^2(\phi_{s,c}^1(\boldsymbol{x})), \phi_{s,c}^2(\phi_{s,c}^1(\boldsymbol{x}')) \rangle$$

$$= \frac{1}{2M} \sum_{i=1}^M \cos(\boldsymbol{\omega}_i^{2\top} (\phi_{s,c}^1(\boldsymbol{x}) - \phi_{s,c}^1(\boldsymbol{x}')))$$

$$\approx \frac{1}{2} \mathbb{E}_{\boldsymbol{\omega}^2 \sim \mathcal{N}(\boldsymbol{0}, \sigma^2 \boldsymbol{I})} [\cos(\boldsymbol{\omega}^{2\top} (\phi_{s,c}^1(\boldsymbol{x}) - \phi_{s,c}^1(\boldsymbol{x}')))] \qquad (7)$$

$$= \frac{1}{2} e^{-\frac{[\|\phi_{s,c}^1(\boldsymbol{x}) - \phi_{s,c}^1(\boldsymbol{x}')\|_2^2 \sigma^2}{2}}$$

$$= \frac{1}{2} [e^{-\frac{[\|\phi_{s,c}^1(\boldsymbol{x})\|_2^2 + \|\phi_{s,c}^1(\boldsymbol{x}')\|_2^2]\sigma^2}{2}}] e^{\sigma^2 \phi_{s,c}^1(\boldsymbol{x})^\top \phi_{s,c}^1(\boldsymbol{x}')},$$

*where $\phi_{s,c}^1(\boldsymbol{x})^\top \phi_{s,c}^1(\boldsymbol{x}') = k_{s,c}^1(\boldsymbol{x}, \boldsymbol{x}')$. By employing the recursive iteration method, we can obtain:*

$$k_{s,c}^L(\boldsymbol{x}, \boldsymbol{x}') = C_{s,c}^L \exp(\delta^2 k_{s,c}^{L-1}(\boldsymbol{x}, \boldsymbol{x}')), L \geq 2,$$

$$C_{s,c}^{L-1} = \frac{1}{2} [e^{-\frac{\|\phi_{s,c}^{L-1} \cdots \phi_{s,c}^1(\boldsymbol{x})\|_2^2 + \|\phi_{s,c}^{L-1} \cdots \phi_{s,c}^1(\boldsymbol{x}')\|_2^2}{2}\sigma^2}]. \qquad (8)$$

*In summary, the proof of Proposition 1 is completed.*

## A.2 Proof of Proposition 2

**Proposition 2** *For the non-stationary spectral kernel under the removal and concatenation scenarios, the spectral kernel mappings are defined by $\phi_{ns,w/o}(\boldsymbol{x}) = [\cos(\boldsymbol{\omega}_1^\top \boldsymbol{x}) + \cos(\boldsymbol{\omega}_1'^\top \boldsymbol{x}), \cdots, \cos(\boldsymbol{\omega}_M^\top \boldsymbol{x}) + \cos(\boldsymbol{\omega}_M'^\top \boldsymbol{x})]^\top$, and $\phi_{ns,c}(\boldsymbol{x}) = [\cos(\boldsymbol{\omega}_1^\top \boldsymbol{x}) + \cos(\boldsymbol{\omega}_1'^\top \boldsymbol{x}), \cdots, \cos(\boldsymbol{\omega}_M^\top \boldsymbol{x}) + \cos(\boldsymbol{\omega}_M'^\top \boldsymbol{x}), \sin(\boldsymbol{\omega}_1^\top \boldsymbol{x}) + \sin(\boldsymbol{\omega}_1'^\top \boldsymbol{x}), \cdots, \sin(\boldsymbol{\omega}_M^\top \boldsymbol{x}) + \sin(\boldsymbol{\omega}_M'^\top \boldsymbol{x})]^\top$, respectively, and $\{\boldsymbol{\omega}_i, \boldsymbol{\omega}_i'\}_{i=1}^M \overset{i.i.d.}{\sim} \mathcal{N}(\boldsymbol{0}, \sigma^2 \boldsymbol{I})$. The stacking of their corresponding deep spectral kernels $k_{ns,w/o}^L(\boldsymbol{x}, \boldsymbol{x}')$ and $k_{ns,c}^L(\boldsymbol{x}, \boldsymbol{x}')$ is formulated as:*

$$k_{ns,w/o}^L(\boldsymbol{x}, \boldsymbol{x}') = C_{ns,w/o}^{L-1}(\cosh(\sigma^2(k_{ns,w/o}^{L-1}(\boldsymbol{x}, \boldsymbol{x}'))) + 2), L \geq 2,$$

$$C_{ns,w/o}^{L-1} = e^{-\frac{\|\phi_{ns,w/o}^{L-1}\cdots\phi_{ns,w/o}^1(\boldsymbol{x})\|_2^2 + \|\phi_{ns,w/o}^{L-1}\cdots\phi_{ns,w/o}^1(\boldsymbol{x}')\|_2^2}{2}\sigma^2}, \tag{9}$$

*and*

$$k_{ns,c}^L(\boldsymbol{x}, \boldsymbol{x}') = C_{ns,c}^{L-1}(\exp(\sigma^2(k_{ns,c}^{L-1}(\boldsymbol{x}, \boldsymbol{x}'))) + 1), L \geq 2,$$

$$C_{ns,c}^{L-1} = 2e^{-\frac{\|\phi_{ns,c}^{L-1}\cdots\phi_{ns,c}^1(\boldsymbol{x})\|_2^2 + \|\phi_{ns,c}^{L-1}\cdots\phi_{ns,c}^1(\boldsymbol{x}')\|_2^2}{2}\sigma^2}. \tag{10}$$

**Proof 2** *Let $k_{ns,w/o}^L(\boldsymbol{x}, \boldsymbol{x}') = \langle \phi_{ns,w/o}^L(\cdots \phi_{ns,w/o}^1(\boldsymbol{x})), \phi_{ns,w/o}^L(\cdots \phi_{ns,w/o}^1(\boldsymbol{x}'))\rangle$ denotes the deep non-stationary spectral kernel with $L$ layers under the removal scenario, where $\phi_{ns,w/o}^1(\boldsymbol{x}) = [\cos(\boldsymbol{\omega}_1^{1\top} \boldsymbol{x}) + \cos(\boldsymbol{\omega}_1'^{1\top} \boldsymbol{x}), \cdots, \cos(\boldsymbol{\omega}_M^{1\top} \boldsymbol{x}) + \cos(\boldsymbol{\omega}_M'^{1\top} \boldsymbol{x})]^\top$ and $\phi_{ns,w/o}^1(\boldsymbol{x}') = [\cos(\boldsymbol{\omega}_1^{1\top} \boldsymbol{x}') + \cos(\boldsymbol{\omega}_1'^{1\top} \boldsymbol{x}'), \cdots, \cos(\boldsymbol{\omega}_M^{1\top} \boldsymbol{x}') + \cos(\boldsymbol{\omega}_M'^{1\top} \boldsymbol{x}')]^\top$. The hierarchical relationship of spectral kernel mapping is formulated by $\phi_{ns,w/o}^L(\boldsymbol{x}) = [\cos(\boldsymbol{\omega}_1^{L\top}(\phi_{ns,w/o}^{L-1}(\boldsymbol{x})) + \cos(\boldsymbol{\omega}_1'^{L\top}(\phi_{ns,w/o}^{L-1}(\boldsymbol{x})), \cdots, \cos(\boldsymbol{\omega}_M^{L\top}(\phi_{s,w/o}^{L-1}(\boldsymbol{x})) + \cos(\boldsymbol{\omega}_M'^{L\top}(\phi_{s,w/o}^{L-1}(\boldsymbol{x}))]^\top$. Following these settings, we have*

$$k_{ns,w/o}^1(\boldsymbol{x}, \boldsymbol{x}') = \langle \phi_{ns,w/o}^1(\boldsymbol{x}), \phi_{ns,w/o}^1(\boldsymbol{x}')\rangle$$

$$= \frac{1}{4M} \sum_{i=1}^M [\cos(\boldsymbol{\omega}_i^{1\top} \boldsymbol{x}) + \cos(\boldsymbol{\omega}_i'^{1\top} \boldsymbol{x})][\cos(\boldsymbol{\omega}_i^{1\top} \boldsymbol{x}') + \cos(\boldsymbol{\omega}_i'^{1\top} \boldsymbol{x}')]$$

$$\approx \frac{1}{4} \mathbb{E}_{\boldsymbol{\omega}^1, \boldsymbol{\omega}'^1 \sim \mathcal{N}(\boldsymbol{0}, \sigma^2 \boldsymbol{I})}[\cos(\boldsymbol{\omega}^{1\top} \boldsymbol{x}) + \cos(\boldsymbol{\omega}'^{1\top} \boldsymbol{x})][\cos(\boldsymbol{\omega}^{1\top} \boldsymbol{x}') + \cos(\boldsymbol{\omega}'^{1\top} \boldsymbol{x}')]$$

$$= \frac{1}{4}\Big[\mathbb{E}_{\boldsymbol{\omega}^1, \boldsymbol{\omega}'^1 \sim \mathcal{N}(\boldsymbol{0}, \sigma^2 \boldsymbol{I})}[\cos(\boldsymbol{\omega}^{1\top} \boldsymbol{x}) \cos(\boldsymbol{\omega}^{1\top} \boldsymbol{x}')] + \mathbb{E}_{\boldsymbol{\omega}^1, \boldsymbol{\omega}'^1 \sim \mathcal{N}(\boldsymbol{0}, \sigma^2 \boldsymbol{I})}[\cos(\boldsymbol{\omega}^{1\top} \boldsymbol{x}) \cos(\boldsymbol{\omega}'^{1\top} \boldsymbol{x}')]$$

$$+ \mathbb{E}_{\boldsymbol{\omega}^1, \boldsymbol{\omega}'^1 \sim \mathcal{N}(\boldsymbol{0}, \sigma^2 \boldsymbol{I})}[\cos(\boldsymbol{\omega}'^{1\top} \boldsymbol{x}) \cos(\boldsymbol{\omega}^{1\top} \boldsymbol{x}')] + \mathbb{E}_{\boldsymbol{\omega}^1, \boldsymbol{\omega}'^1 \sim \mathcal{N}(\boldsymbol{0}, \sigma^2 \boldsymbol{I})}[\cos(\boldsymbol{\omega}'^{1\top} \boldsymbol{x}) \cos(\boldsymbol{\omega}'^{1\top} \boldsymbol{x}')]\Big]$$

$$= e^{-\frac{[\|\boldsymbol{x}\|_2^2 + \|\boldsymbol{x}'\|_2^2]\sigma^2}{2}} \cosh(\sigma^2 \boldsymbol{x}^\top \boldsymbol{x}'). \tag{11}$$

*Then, the deep non-stationary spectral kernel with $L = 2$ can be formulated as follows:*

$$k_{ns,w/o}^2(\boldsymbol{x}, \boldsymbol{x}') = \langle \phi_{ns,w/o}^2(\phi_{ns,w/o}^1(\boldsymbol{x})), \phi_{ns,w/o}^2(\phi_{ns,w/o}^1(\boldsymbol{x}'))\rangle$$

$$= \frac{1}{4M} \sum_{i=1}^M [\cos(\boldsymbol{\omega}_i^{2\top} \phi_{ns,w/o}^1(\boldsymbol{x})) + \cos(\boldsymbol{\omega}_i'^{2\top} \phi_{ns,w/o}^1(\boldsymbol{x}))][\cos(\boldsymbol{\omega}_i^{2\top} \phi_{ns,w/o}^1(\boldsymbol{x}')) + \cos(\boldsymbol{\omega}_i'^{2\top} \phi_{ns,w/o}^1(\boldsymbol{x}'))]$$

$$\approx \frac{1}{4} \mathbb{E}_{\boldsymbol{\omega}^2, \boldsymbol{\omega}'^2 \sim \mathcal{N}(\boldsymbol{0}, \sigma^2 \boldsymbol{I})}[\cos(\boldsymbol{\omega}_i^{2\top} \phi_{ns,w/o}^1(\boldsymbol{x})) + \cos(\boldsymbol{\omega}_i'^{2\top} \phi_{ns,w/o}^1(\boldsymbol{x}))][\cos(\boldsymbol{\omega}_i^{2\top} \phi_{ns,w/o}^1(\boldsymbol{x}')) + \cos(\boldsymbol{\omega}_i'^{2\top} \phi_{ns,w/o}^1(\boldsymbol{x}'))]$$

$$= e^{-\frac{[\|\phi_{ns,w/o}^1(\boldsymbol{x})\|_2^2 + \|\phi_{ns,w/o}^1(\boldsymbol{x}')\|_2^2]\sigma^2}{2}} \cosh(\sigma^2 \phi_{ns,w/o}^1(\boldsymbol{x})^\top \phi_{ns,w/o}^1(\boldsymbol{x}')) \tag{12}$$

*By employing the recursive iteration method, we can obtain:*

$$k_{ns,w/o}^L(\boldsymbol{x}, \boldsymbol{x}') = C_{ns,w/o}^{L-1}(\cosh(\sigma^2(k_{ns,w/o}^{L-1}(\boldsymbol{x}, \boldsymbol{x}'))) + 2), L \geq 2,$$

$$C_{ns,w/o}^{L-1} = e^{-\frac{\|\phi_{ns,w/o}^{L-1}\cdots\phi_{ns,w/o}^1(\boldsymbol{x})\|_2^2 + \|\phi_{ns,w/o}^{L-1}\cdots\phi_{ns,w/o}^1(\boldsymbol{x}')\|_2^2}{2}\sigma^2}. \tag{13}$$

*Similar to the removal scenario, let $k_{ns,c}^L(\boldsymbol{x}, \boldsymbol{x}') = \langle \phi_{ns,c}^L(\cdots \phi_{ns,c}^1(\boldsymbol{x})), \phi_{ns,c}^L(\cdots \phi_{ns,c}^1(\boldsymbol{x}'))\rangle$ denotes the deep non-stationary spectral kernel with $L$ layers under the concatenation scenario,*

where $\phi^1_{ns,c}(\boldsymbol{x}) = [\cos(\boldsymbol{\omega}_1^{1\top}\boldsymbol{x}) + \cos(\boldsymbol{\omega}_1^{'1\top}\boldsymbol{x}), \cdots, \cos(\boldsymbol{\omega}_M^{1\top}\boldsymbol{x}) + \cos(\boldsymbol{\omega}_M^{'1\top}\boldsymbol{x}), \sin(\boldsymbol{\omega}_1^{1\top}\boldsymbol{x}) + \sin(\boldsymbol{\omega}_1^{'1\top}\boldsymbol{x}), \cdots, \sin(\boldsymbol{\omega}_M^{1\top}\boldsymbol{x}) + \sin(\boldsymbol{\omega}_M^{'1\top}\boldsymbol{x})]^\top$ and $\phi^1_{ns,c}(\boldsymbol{x}') = [\cos(\boldsymbol{\omega}_1^{1\top}\boldsymbol{x}') + \cos(\boldsymbol{\omega}_1^{'1\top}\boldsymbol{x}'), \cdots, \cos(\boldsymbol{\omega}_M^{1\top}\boldsymbol{x}') + \cos(\boldsymbol{\omega}_M^{'1\top}\boldsymbol{x}'), \sin(\boldsymbol{\omega}_1^{1\top}\boldsymbol{x}') + \sin(\boldsymbol{\omega}_1^{'1\top}\boldsymbol{x}'), \cdots, \sin(\boldsymbol{\omega}_M^{1\top}\boldsymbol{x}') + \sin(\boldsymbol{\omega}_M^{'1\top}\boldsymbol{x}')]^\top$. *The hierarchical relationship of spectral kernel mapping is formulated by* $\phi^L_{ns,c}(\boldsymbol{x}) = [\cos(\boldsymbol{\omega}_1^{L\top}(\phi^{L-1}_{ns,c}(\boldsymbol{x})) + \cos(\boldsymbol{\omega}_1^{'L\top}(\phi^{L-1}_{s,c}(\boldsymbol{x})), \cdots, \cos(\boldsymbol{\omega}_M^{L\top}(\phi^{L-1}_{s,c}(\boldsymbol{x})) + \cos(\boldsymbol{\omega}_M^{'L\top}(\phi^{L-1}_{s,c}(\boldsymbol{x})), \sin(\boldsymbol{\omega}_1^{L\top}(\phi^{L-1}_{ns,c}(\boldsymbol{x})) + \sin(\boldsymbol{\omega}_1^{'L\top}(\phi^{L-1}_{ns,c}(\boldsymbol{x})), \cdots, \sin(\boldsymbol{\omega}_M^{L\top}(\phi^{L-1}_{s,c}(\boldsymbol{x})) + \sin(\boldsymbol{\omega}_M^{'L\top}(\phi^{L-1}_{s,c}(\boldsymbol{x}))]^\top$. *Following these settings, we have*

$$k^1_{ns,c}(\boldsymbol{x}, \boldsymbol{x}') = \langle \phi^1_{ns,c}(\boldsymbol{x}), \phi^1_{ns,c}(\boldsymbol{x}') \rangle = \langle \phi^1_{ns,c}(\boldsymbol{x}), \phi^1_{ns,c}(\boldsymbol{x}') \rangle$$

$$= \frac{1}{4M}\sum_{i=1}^{M}[\cos(\boldsymbol{\omega}_i^{1\top}\boldsymbol{x}) + \cos(\boldsymbol{\omega}_i^{'1\top}\boldsymbol{x})][\cos(\boldsymbol{\omega}_i^{1\top}\boldsymbol{x}') + \cos(\boldsymbol{\omega}_i^{'1\top}\boldsymbol{x}')] + [\sin(\boldsymbol{\omega}_i^{1\top}\boldsymbol{x}) + \sin(\boldsymbol{\omega}_i^{'1\top}\boldsymbol{x})][\sin(\boldsymbol{\omega}_i^{1\top}\boldsymbol{x}') + \sin(\boldsymbol{\omega}_i^{'1\top}\boldsymbol{x}')]$$

$$\approx \frac{1}{4}\mathbb{E}_{\boldsymbol{\omega}^1,\boldsymbol{\omega}^{'1}\sim\mathcal{N}(\boldsymbol{0},\sigma^2\boldsymbol{I})}[\cos(\boldsymbol{\omega}_i^{1\top}\boldsymbol{x}) + \cos(\boldsymbol{\omega}_i^{'1\top}\boldsymbol{x})][\cos(\boldsymbol{\omega}_i^{1\top}\boldsymbol{x}') + \cos(\boldsymbol{\omega}_i^{'1\top}\boldsymbol{x}')] + [\sin(\boldsymbol{\omega}_i^{1\top}\boldsymbol{x}) + \sin(\boldsymbol{\omega}_i^{'1\top}\boldsymbol{x})][\sin(\boldsymbol{\omega}_i^{1\top}\boldsymbol{x}') + \sin(\boldsymbol{\omega}_i^{'1\top}\boldsymbol{x}')].$$

$$(14)$$

*Based on all the above results, we can obtain:*

$$k^1_{ns,c}(\boldsymbol{x}, \boldsymbol{x}') = 2e^{-\frac{[\|\boldsymbol{x}\|_2^2 + \|\boldsymbol{x}'\|_2^2]\sigma^2}{2}}\exp(\sigma^2\boldsymbol{x}^\top\boldsymbol{x}'), \quad (15)$$

*and the case of $L = 2$ can be formulated as follows:*

$$k^2_{ns,c}(\boldsymbol{x}, \boldsymbol{x}') = \langle \phi^2_{ns,c}(\phi^1_{ns,c}(\boldsymbol{x})), \phi^2_{ns,c}(\phi^1_{ns,c}(\boldsymbol{x}')) \rangle$$

$$\approx 2e^{-\frac{[\|\phi^1_{ns,c}(\boldsymbol{x})\|_2^2 + \|\phi^1_{ns,c}(\boldsymbol{x}')\|_2^2]\sigma^2}{2}}\exp(\sigma^2\phi^1_{ns,c}(\boldsymbol{x})^\top\phi^1_{ns,c}(\boldsymbol{x}')). \quad (16)$$

*By employing the recursive iteration method, we can obtain:*

$$k^L_{ns,c}(\boldsymbol{x}, \boldsymbol{x}') = C^{L-1}_{ns,c}(\exp(\sigma^2(k^{L-1}_{ns,c}(\boldsymbol{x}, \boldsymbol{x}'))) + 1), L \geq 2,$$

$$C^{L-1}_{ns,c} = 2e^{-\frac{\|\phi^{L-1}_{ns,c}\cdots\phi^1_{ns,c}(\boldsymbol{x})\|_2^2 + \|\phi^{L-1}_{ns,c}\cdots\phi^1_{ns,c}(\boldsymbol{x}')\|_2^2}{2}\sigma^2}. \quad (17)$$

*In summary, the proof of Proposition 2 is completed.*

## B  DEEP GAUSSIAN KERNEL BASED ON BOCHNER'S THEOREM

**Lemma 1** *Huang et al. (2023) Given the Gaussian kernel* $k^0 = \exp(-\lambda\|\boldsymbol{x} - \boldsymbol{x}'\|^2) = \int_{\mathbb{R}^d} e^{-i(\boldsymbol{x}-\boldsymbol{x}')\boldsymbol{\omega}}s^0(\boldsymbol{\omega})d\boldsymbol{\omega}$, *and* $s^0(\boldsymbol{\omega}) = \frac{1}{(2\sqrt{\lambda\pi})^d}\exp(-\frac{\|\boldsymbol{\omega}\|^2}{4\lambda})$ *with* $\boldsymbol{x}, \boldsymbol{x}', \boldsymbol{\omega} \in \mathbb{R}^d, \lambda > 0$. *The deep Gaussian kernel defined by* $k^L(\boldsymbol{x}, \boldsymbol{x}') = \exp(k^{L-1}(\boldsymbol{x}, \boldsymbol{x}'))$, *it holds*

$$k^L(\boldsymbol{x}, \boldsymbol{x}') = e^L(0)\sum_{p=0}^{\infty}\frac{\beta_{L,p}}{p!}\exp(-p\lambda\|\boldsymbol{x} - \boldsymbol{x}'\|_2^2), p \in \mathbb{N},$$

$$k^L(\boldsymbol{x}, \boldsymbol{x}') = e^L(0) + h^L(\boldsymbol{x}, \boldsymbol{x}'), \quad h^L(\boldsymbol{x}, \boldsymbol{x}') = \int_{\mathbb{R}^d} e^{-i(\boldsymbol{x}-\boldsymbol{x}')\cdot\boldsymbol{\omega}}s^L(\boldsymbol{\omega})d\boldsymbol{\omega}, \quad (18)$$

$$s^L(\boldsymbol{\omega}) = \frac{e^L(0)}{(2\sqrt{\lambda\pi})^d}\sum_{p=1}^{\infty}\frac{\beta_{L,p}}{L!L^{d/2}}\exp(-\frac{\|\boldsymbol{\omega}\|_2^2}{4\lambda L}).$$

**Proof 3** *The proof can be found in Huang et al. (2023).*

Lemma 1 indicates that the deep Gaussian kernel tends to be a combination of Gaussian kernels with different bandwidth parameters. Furthermore, the bandwidth parameter $p\lambda$ depends solely on $\lambda$, since $p$ is a given number for each Gaussian component. The weight of each Gaussian component is given by $\frac{\beta_{L,p}}{p!}$. These observations suggest that, for fixed $\lambda$ and $L$, the deep Gaussian kernel can be regarded as a combination of multiple Gaussian kernels with determined components. As the depth increases, the constituent components remain unchanged, while only their corresponding weighting coefficients are adaptively adjusted. By considering the first five Gaussian kernel components, we

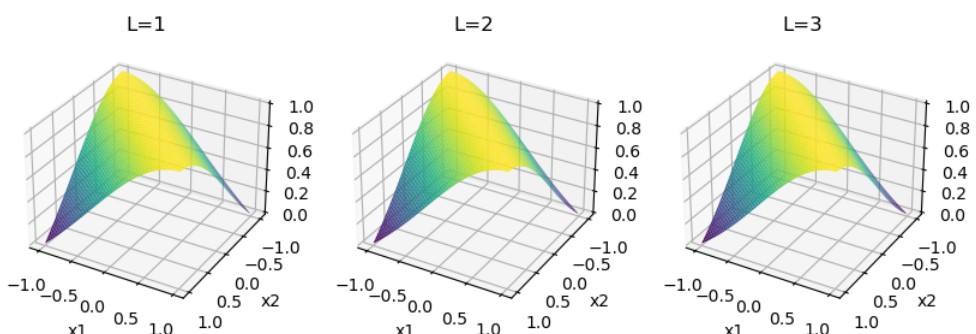

Figure 1: The deep Gaussian kernel with $L = 1, 2, 3$.

illustrate the deep Gaussian kernel with $L = 1, 2, 3$ in Figure 1, which is consistent with the afore-mentioned analysis. The corresponding weights associated with each component are shown in Table 1.

From the spectral kernel perspective, the evolution of the deep Gaussian kernel is primarily governed by changes in its associated spectral density. Such spectral density functions also tend to include determined components (*i.e.,* Gaussian spectral density) with the given $\lambda$ and $L$. This means that we can construct the deep spectral kernel by directly resampling from an adaptive spectral density.

Table 1: The weight of Gaussian component within the deep Gaussian kernel.

| $\frac{\beta_{L,p}}{p!p^{1/2}}$ | $L = 1$ | $L = 2$ | $L = 3$ |
|---|---|---|---|
| $p = 1$ | $\beta_{1,1}=1, \frac{\beta_{L,p}}{p!p^{1/2}}=1, \frac{\beta_{L,p}}{p!}=1$ | $\beta_{2,1}=1, \frac{\beta_{L,p}}{p!p^{1/2}}=1, \frac{\beta_{L,p}}{p!}=1$ | $\beta_{3,1}=1, \frac{\beta_{L,p}}{p!p^{1/2}}=1, \frac{\beta_{L,p}}{p!}=1$ |
| $p = 2$ | $\beta_{1,2}=2, \frac{\beta_{L,p}}{p!p^{1/2}}=\frac{2}{2\sqrt{2}}, \frac{\beta_{L,p}}{p!}=1$ | $\beta_{2,2}=1, \frac{\beta_{L,p}}{p!p^{1/2}}=\frac{1}{2\sqrt{2}}, \frac{\beta_{L,p}}{p!}=\frac{1}{2}$ | $\beta_{3,2}=1, \frac{\beta_{L,p}}{p!p^{1/2}}=\frac{1}{2\sqrt{2}}, \frac{\beta_{L,p}}{p!}=\frac{1}{2}$ |
| $p = 3$ | $\beta_{1,3}=5, \frac{\beta_{L,p}}{p!p^{1/2}}=\frac{5}{6\sqrt{3}}, \frac{\beta_{L,p}}{p!}=\frac{5}{6}$ | $\beta_{2,3}=3, \frac{\beta_{L,p}}{p!p^{1/2}}=\frac{3}{6\sqrt{3}}, \frac{\beta_{L,p}}{p!}=\frac{3}{6}$ | $\beta_{3,3}=1, \frac{\beta_{L,p}}{p!p^{1/2}}=\frac{1}{6\sqrt{3}}, \frac{\beta_{L,p}}{p!}=\frac{1}{6}$ |
| $p = 4$ | $\beta_{1,4}=15, \frac{\beta_{L,p}}{p!p^{1/2}}=\frac{15}{24\sqrt{4}}, \frac{\beta_{L,p}}{p!}=\frac{15}{24}$ | $\beta_{2,4}=7, \frac{\beta_{L,p}}{p!p^{1/2}}=\frac{7}{24\sqrt{4}}, \frac{\beta_{L,p}}{p!}=\frac{7}{24}$ | $\beta_{3,4}=4, \frac{\beta_{L,p}}{p!p^{1/2}}=\frac{4}{24\sqrt{4}}, \frac{\beta_{L,p}}{p!}=\frac{4}{24}$ |
| $p = 5$ | $\beta_{1,5}=52, \frac{\beta_{L,p}}{p!p^{1/2}}=\frac{52}{120\sqrt{5}}, \frac{\beta_{L,p}}{p!}=\frac{52}{120}$ | $\beta_{2,5}=15, \frac{\beta_{L,p}}{p!p^{1/2}}=\frac{15}{120\sqrt{5}}, \frac{\beta_{L,p}}{p!}=\frac{15}{120}$ | $\beta_{3,5}=11, \frac{\beta_{L,p}}{p!p^{1/2}}=\frac{11}{120\sqrt{5}}, \frac{\beta_{L,p}}{p!}=\frac{11}{120}$ |

## C  THE COMPARED METHODS

To further evaluate the proposed generative spectral kernel, we compare several SOTA deep spectral kernel methods, including **TRF** Shilton et al. (2022), **GRFF** Fang et al. (2023), and **DKEF** Wenliang et al. (2019) for the stationary case, as well as **DSKN** Xue et al. (2019), **ASKL** Li et al. (2020), and **CokeNet** Tian et al. (2024) for the non-stationary case.

- **TRF**: Tuned Random Features, which selects the spectral density function from a repro-ducing kernel Hilbert space to search the space of all stationary kernels.

- **GRFF**: Generative Random Fourier Features, an end-to-end spectral kernel learning ap-proach that models the spectral density distribution of the stationary kernel via a generative network based on the random Fourier features. It is worth noting that this method is similar to our plain deep stationary spectral kernel. The key distinction lies in the fact that the architecture of our plain deep stationary spectral kernel is constructed in the concatenation scenario.

- **DKEF**: Deep Kernels for Exponential Family Densities, a scheme for learning a kernel parameterized by a deep network. This gives a very rich class of density models, capable of fitting complex structures on moderate-dimensional problems.

- **DSKN**: Deep Spectral Kernel Network, which integrates the non-stationary spectral kernel into the deep architecture. This method models deep spectral kernels by stacking the non-stationary spectral kernel mappings.

- **ASKL**: Automated Spectral Kernel Learning, an efficient learning framework that incor-porates the process of finding suitable kernels and model training. This method introduced

regularization terms via investigating the effect of the derived data-dependent generalization error bounds.

- **CokeNet**: Copula-Nested Spectral Kernel Network, the core idea of this method is to introduce copula networks into the design of the spectral density based on Sklar's theorem to extend the range of hypothesis space of deep spectral kernels.

# D  ADDITIONAL EXPERIMENT RESULTS

In this section, we report the additional experiment results, including the data distribution and loss landscape of the other datasets.

## D.1  DATA DISTRIBUTION

For completeness, we also provide the results of the time series classification task in Table 2. The data distributions of the remaining datasets are visualized in Figure 2. As discussed in Section 5.2.1, our proposed generative spectral kernel exhibits advantages in capturing the inherent patterns within the data. For datasets that are well-separated and exhibit high intra-class compactness, our proposed method achieves performance comparable to that of other methods. For example, the similar performance (93.59% vs. 93.40%) on **ECG5000** and (91.70% vs. 92.00%) on **ECG200** compared with the mainstream deep spectral kernel methods. By contrast, for datasets lacking clear class separability, our proposed generative spectral kernel outperforms other methods impressively and achieves 11.35% accuracy increment (65.20 → 72.60) on **Rock** and 7.79% (76.11 → 82.04) on **FordA**. These results indicate the effectiveness of our method in uncovering the essential characteristics of the data.

Table 2: Time series classification results (accuracy (Acc) and parameters (Para)) for the stationary (S) and non-stationary (Ns) cases. The best results are highlighted in **bold**.

| | Model | FordA | | ECG200 | | ECG5000 | | Rock | | Trace | | Plane | |
|---|---|---|---|---|---|---|---|---|---|---|---|---|---|
| | | Acc | Para | Acc | Para | Acc | Para | Acc | Para | Acc | Para | Acc | Para |
| | plain | 74.28 | 1.59M | 90.70 | 1.38M | 93.05 | 1.40M | 64.80 | 2.79M | 81.70 | 1.47M | 95.81 | 1.41M |
| | TRF | 65.74 | 0.52M | 88.60 | 0.32M | 91.72 | 0.34M | 59.60 | 1.72M | 74.90 | 0.41M | 94.57 | 0.34M |
| S | GRFF | 74.92 | **0.28M** | 85.90 | **0.23M** | 93.10 | **0.24M** | 69.60 | **0.59M** | 81.70 | **0.25M** | 96.76 | **0.24M** |
| | DKEF | 76.65 | 0.65M | 87.80 | 0.37M | 91.94 | 0.40M | 59.20 | 1.81M | 75.70 | 0.47M | 96.38 | 0.39M |
| | Ours | **79.42** | 0.43M | **91.70** | 0.32M | **93.14** | 0.33M | **71.40** | 1.03M | **82.90** | 0.37M | **96.89** | 0.34M |
| | plain | 76.11 | 2.89M | **92.00** | 2.48M | 93.09 | 2.53M | 65.20 | 5.29M | 82.10 | 2.66M | 95.71 | 2.53M |
| | DSKN | 75.47 | 1.72M | 90.00 | 1.30M | 92.78 | 1.35M | 59.60 | 4.12M | 81.70 | 1.49M | 94.95 | 1.35M |
| Ns | ASKL | 72.50 | 2.00M | 87.53 | **0.39M** | 92.75 | 0.57M | 44.00 | 5.70M | 80.00 | 1.11M | 97.14 | 0.59M |
| | CokeNet | 71.61 | 1.58M | 89.95 | 0.76M | 93.40 | 0.85M | 57.50 | 6.39M | 76.05 | 1.12M | 97.14 | 0.86M |
| | Ours | **82.04** | **0.79M** | 91.70 | 0.42M | **93.59** | 0.46M | **72.60** | 3.31M | **83.60** | 0.58M | **97.43** | 0.46M |

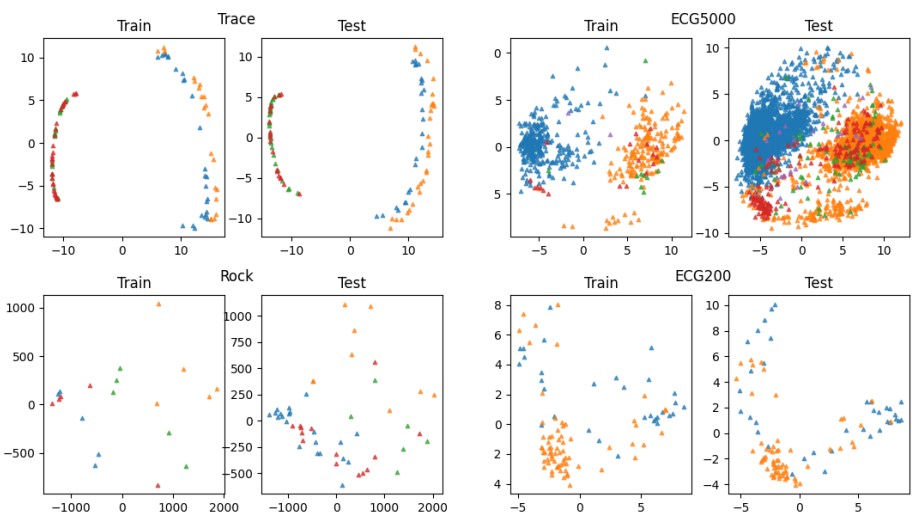

Figure 2: The data distribution of the remaining datasets

### D.2 LOSS LANDSCAPE

The spectral kernel mapping can be interpreted as a single-layer neural network equipped with a periodic activation function. The stacking will lead to a highly non-convex loss landscape with numerous local minima, thereby posing a significant optimization challenge Xue & Wu (2020). Compared with deep spectral kernel models, the proposed generative spectral kernel is constructed with a single-layer spectral kernel architecture, thereby circumventing the associated optimization challenges. To evaluate this advantage, we compare the loss landscape for the deep spectral kernel model and the generative spectral kernel model on the time series classification task.

In this experiment, the proposed generative spectral kernel follows the same setting as described in Section 5.2.1. The generators ($G_1$ and $G_2$) and encoders ($B_1$ and $B_2$) are simultaneously designed as a two-layer fully connected network with ReLU activation. $\Upsilon_1$ and $\Upsilon_2$ are set as the linear transformation. All models are trained using the ADAM algorithm with a learning rate of 0.01.

We visualize the loss landscape of the remaining datasets in Figure 3. We can observe that the loss landscape of the deep spectral kernel model is rugged with a sharp minimum point in most instances.

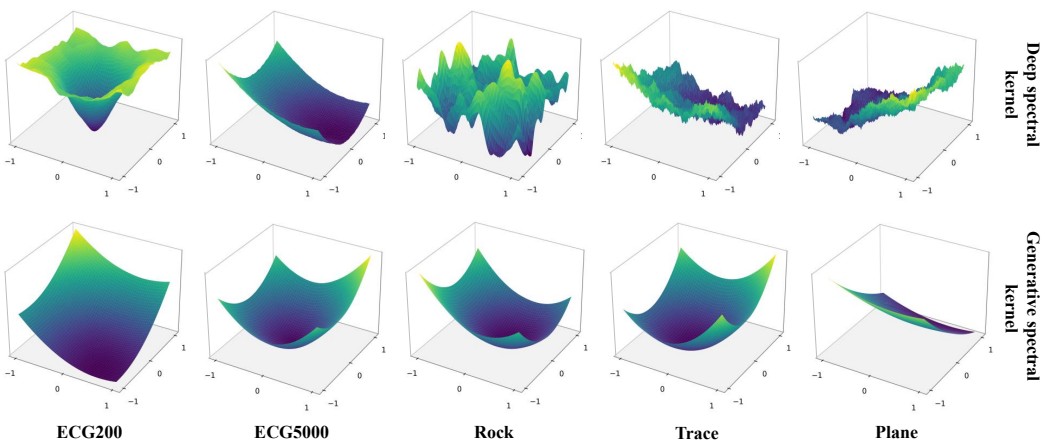

Figure 3: The loss landscape of the remaining datasets

### D.3 THE ADAPTIVE SPECTRAL DENSITY

To further assess the effectiveness of the proposed GensKer in generating adaptive spectral densities, the kernel regression task (the same as Section 5.1.1 of the main text) is performed on the synthetic data for the deep spectral kernel and the proposed generative spectral kernel. Specifically, the deep spectral kernel ($L = 1, 2, 3$, respectively) is constructed by stacking the non-stationary spectral kernel mappings under the concatenation scenario. The architecture of the generative spectral kernel is the same as that in Section 5.2.1. Since the generative spectral kernel is not equipped with a deep architecture, the same model is applied to the cases of $L = 1, 2, 3$.

We first elaborate a series of synthetic data $\{x_i, y_i\}_{i=1}^{6000}$ by the function $y_i = \sin(x_i) + \sin(3x_i) + \sin(5x_i) + \sin(10x_i), x_i \in [0, 10]$, where 5000 samples are used for training and 1000 for testing. Then, we evaluate their performance on the kernel regression task. The result is shown in Figure 4. It demonstrates the following information: (1) The proposed generative spectral kernel consistently shows outstanding performance on $L = 1, 2, 3$, which is attributed to the powerful representation ability of the generated spectral density. (2) The deep spectral kernel underperforms over $x \in [4, 6]$, but its performance improves progressively as the number of layers increases. In particular, under $L = 3$, the performance of the deep spectral kernel is nearly comparable to that of the generative spectral kernel. This observation further indicates that the RKHS induced by the deep spectral

kernel in the concatenation setting undergoes progressive expansion with increasing depth, thereby contributing to performance gains.

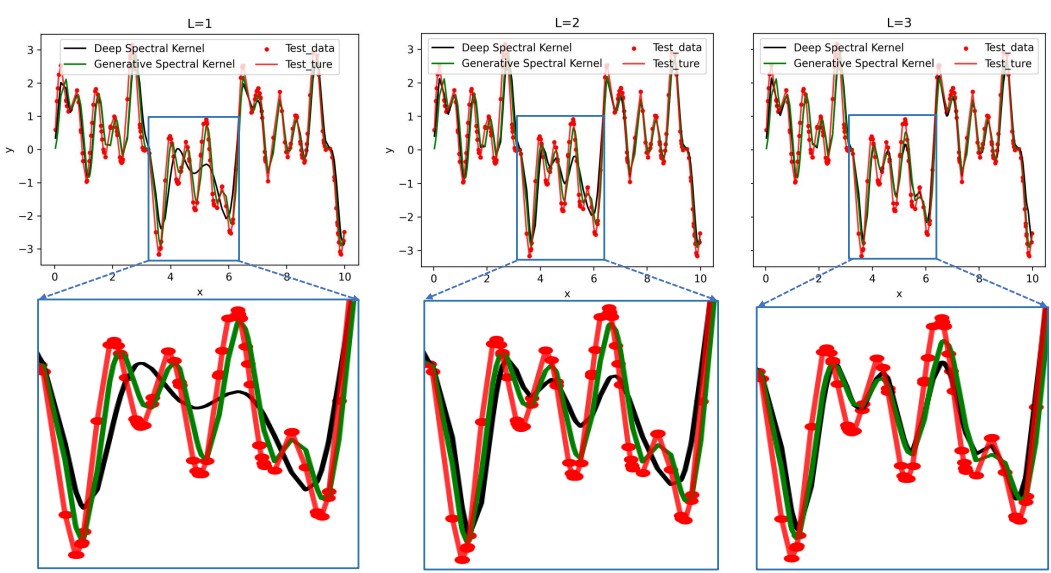

Figure 4: The regression results of different models.

## E    THE STATEMENT OF THE USE OF LARGE LANGUAGE MODELS (LLMS)

We declare that the large language model (LLM) was used solely for language polishing (except for the definition, lemma, theorem, and proposition), including the grammar, wording, and readability.

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
