# OpenReview forum: "GensKer: Generative Spectral Kernel with RKHS Expansion Guarantees"
_ICLR.cc/2026/Conference — Submitted to ICLR 2026_

### Official Review · Reviewer_pJsy · 2025-10-22

**Soundness:** 3
**Presentation:** 2
**Contribution:** 3
**Rating:** 6
**Confidence:** 3

**Summary:**

The authors propose a deep spectral kernel learning framework, called GensKer. The framework consists of two modules, spectral generative module and spectral kernel module. These modules are based on the spectral properties of kernels. They also theoretically show the expansion of the RKHS by considering the deep structure of the proposed kernels.

**Strengths:**

The proposed framework is based on the theoretical peoperties of the kernels. Also, they theoretically investigate the growth of the representation power by considering the deep structure of kernels and confirm it empirically. The topic is interesting and the theoretical investigations are solid.

**Weaknesses:**

- In equations 6-9, the constants $C_{s,w/o}^{L-1}, C_{s,c}^{L-1}, C_{ns,w/o}^{L-1}, C_{ns,c}^{L-1}$ depend on $x$. Does this mean $C_{s,w/o}^{L-1}, C_{s,c}^{L-1}, C_{ns,w/o}^{L-1}, C_{ns,c}^{L-1}$ are functions of $x$, or do they depend on training samples? It was not clear for me the rigorous meaning of "data-dependent spectral density" in line 268. Could you clarify that? In addition, could you clarify why the proposed architecture enables GensKer to circumvent the optimization challenges posed by deep spectral kernels?

- In section 3.2, the authors show that the RKHSs expands as the number $L$ of layers increases. Can you show the universality of the RKHS with sufficiently large $L$? Also, for the case of neural networks, the growth of the representation power is exponential in some sense (e.g. Cohen et al., JMLR, 2016). Can you evaluate the speed of the growth of the representation power for the proposed kernel?

- The the references should be cited with parenthesis (use ```\citep```).

**Minor comments:**
- l 218: By and Lemma 3 shoud be By Lemma 3?

**Questions:**

Please see the weakness section.

---

> ### Author Response · Authors · 2025-11-21
> **Response to Reviewer pJsy**
>
> **W1**：$C_{s, w/o}^{L-1}, C_{s, c}^{L-1}, C_{ns, w/o}^{L-1}, C_{ns, c}^{L-1}$ are constants, even though they appear to depend on $\boldsymbol{x}$.
>
> For the stationary spectral kernels, using $C_{s, w/o}^{L-1} $ as an example, within its formulation $||\phi_ {s, w/o}^{L-1}\cdots \phi_ {s, w/o}^1(\boldsymbol{x})||_ 2^2 = \langle \phi_{s, w/o}^{L-1}\cdots \phi_{s, w/o}^1(\boldsymbol{x}), \phi_{s, w/o}^{L-1}\cdots \phi_{s, w/o}^1(\boldsymbol{x})\rangle = k^{L-1}(\boldsymbol{x},\boldsymbol{x}) =1$ based on Equation 1. Therefore, $C_{s, w/o}^{L-1} = \frac{1}{2}[e^{-\frac{1+1}{2}\sigma^2}] = \frac{1}{2}e^{-\sigma^2}$ is a constant.
>
> For the non-stationary spectral kernels, Equation (2) holds under relatively strict conditions, which require the measure $\mu$ is the Lebesgue-Stieltjes measure associated with some positive semi-definite spectral density function $s(\boldsymbol{\omega}, \boldsymbol{\omega}')$ with bounded variation *i.e.,* $\int_{\mathbb{R}^d \times \mathbb{R}^d} s(\boldsymbol{\omega}, \boldsymbol{\omega}') < \infty$ and satisfies $\mu(d\boldsymbol{\omega},d\boldsymbol{\omega}') = s(\boldsymbol{\omega},\boldsymbol{\omega}') d\boldsymbol{\omega}d\boldsymbol{\omega}'$. Furthermore, the explicit spectral kernel mapping is obtained based on the assumption that the Lebesgue-Stieltjes measure with a right-continuous monotonically increasing distribution function $F_{\mu}:\mathbb{R}^d \times \mathbb{R}^d \to \mathbb{R}$, i.e., $\mu(a, b] = F_{\mu}(b)-F_{\mu}(a)$. Therefore, $s(\boldsymbol{\omega}, \boldsymbol{\omega}') \geq 0$ almost everywhere and is diagonally dominant. Using $C_{ns, w/o}^{L-1} $ as an example, within its formulation, $||\phi_ {s, w/o}^{L-1}\cdots \phi_ {s, w/o}^1(\boldsymbol{x})||_ 2^2 = \langle \phi_{s, w/o}^{L-1}\cdots \phi_{s, w/o}^1(\boldsymbol{x}), \phi_{s, w/o}^{L-1}\cdots \phi_{s, w/o}^1(\boldsymbol{x})\rangle = k^{L-1}(\boldsymbol{x},\boldsymbol{x}) = \int_{\mathbb{R}^d}s(\boldsymbol{\omega}, \boldsymbol{\omega}) d\boldsymbol{\omega}$. Thereby, $C_{ns, w/o}^{L-1} = \frac{1}{2}[e^{-\frac{c_{\mu}+c_{\mu}}{2}\sigma^2}] = \frac{1}{2}e^{-c_{\mu}\sigma^2}$ is a constant, where $c_{\mu}$ is constant relative to the measure $\mu$.
>
> As a result, we treat $C_{s, w/o}^{L-1}, C_{s, c}^{L-1}, C_{ns, w/o}^{L-1}, C_{ns, c}^{L-1}$  as constants. We have provided a more detailed discussion in the revised manuscript (Appendix A).
>
> **data-dependent spectral density:** "data-dependent spectral density" means that the spectral density is adaptively learned from the data. Based on Bochner's theorem and Yaglom's theorem, there exists a one-to-one correspondence between the positive semi-definite kernel and the spectral density. This indicates that we can learn a kernel through learn an adaptive spectral density. In GensKer, we first generate the spectral density $s(\boldsymbol{\omega}, \boldsymbol{\omega}')$ using the designed spectral generation module and then construct the spectral kernel based on the generated spectral density. Finally, the constructed spectral kernel is used to perform the task. Remarkably, all these processes are end-to-end. Therefore, we can learn a data-dependent spectral density.
>
> **Optimization challenges:** The primary reason is that the proposed GensKer incorporates only one spectral kernel mapping layer. In the spectral kernel perspective, the progressive expansion of RKHSs induced by the deep spectral kernels is linked to their corresponding spectral densities. This means we can implement the deep spectral kernel by directly resampling from an adaptive spectral density, without a deep architecture. The proposed GensKer adaptively generates the spectral density from the data and subsequently constructs the spectral kernel via incorporating only one spectral kernel mapping layer. Therefore, Gensker can circumvent the optimization challenges introduced by multi-layer stacking in deep spectral kernels.
>
> **W3 and Minor Comments:** Thank you for your careful review of our manuscript. We have corrected these points in the revised manuscript.

---

> ### Author Response · Authors · 2025-11-21
> **Response to Reviewer pJsy**
>
> **W2:**
>
> **The universality of the RKHS:** For a deep spectral kernel constructed by stacking the non-stationary spectral kernel mapping under the concatenation case, the RKHS associated with this deep spectral kernel can approximate any $2\pi$-periodic continuous function $f\in \Psi$ with sufficiently large $L$. $\Psi$ denotes the $2\pi$-periodic continuous function space.
>
> Before stating the conclusion, we introduce two definitions.
>
> **Definition 1 (Function Modulus of Smoothness)**. Let $f\in \Psi$ and $\delta \geq 0$. The first-order modulus of smoothness of $f$ in the $\Psi$-norm, denote by $\omega(f, \delta)_{\Psi}$, is defined as :
>
> $$
> \omega(f, \delta)_ {\Psi} = \sup_ {\{||\Delta||_ 2 \leq \delta\}} \{ ||f(x + \Delta) - f(x)||_\Psi, \quad \forall \Delta \in \mathbb{R}^d \}.
> $$
>
> Furthermore, the following inequality holds:
>
> $$
> \omega(f, \lambda \delta)_ {\Psi} \leq (1 + \lambda) \omega(f, \delta)_{\Psi}, \quad \forall \lambda > 0.
> $$
>
> **Definition 2 (Composite Function)**.  Let $\mathcal{G}$ be a directed acyclic graph with a set of source nodes $S$ and a set of vertices $V$. Let $d_v$ be the in-degree of $v \in V$. Let $f$ be a composite $\mathcal{G}$-function that shares the same compositional graph structure with $\mathcal{G}$, *i.e.,* $f$ can be recursively constructed by the composition of a set of constituent functions $(f_v)_ { v \in V }$ layer by layer. The input to a constituent function $f_v$ at an intermediate layer is the set of constituent functions from the previous layer $(f_{v_ i} )_ {i=1}^{d_v}$, corresponding to the node $v$ and its predecessor nodes $(v_i )_{i=1}^{d_v}$ in $\mathcal{G}$. Thus, the structure of the composite function $f$ corresponds to $\mathcal{G}$ with $|S|$-dimensional input.
>
> Let $f\in \Psi$ be a composite $\mathcal{G}$-function consisting of a set of constituent functions $(f_v \in \Psi)_ {v\in V}$. Let $\mathcal{K}$ be isomorphic to $f$ and $\mathcal{G}$, *i.e.,* it shares the same directed acyclic graph network structure. For each node $v \in V$, let $\mathcal{K}_ v^{(1)} = \sum_{i=1}^{M_v} \phi_i$ correspond to $v$. The network width of $\mathcal{K}_v^{(1)}$ is $M_v=(\lambda_v+1)^{d_v}-1$. The approximation error under the
> $\Psi$-norm can be estimated by:
>
> $$
> \inf_ {\mathcal{K}} \sup_ {f}  ||\mathcal{K} - f||_ {\Psi} \leq \sum_ {v \in V} \[D_{v} \omega_{\lambda_v} + (1 + (\lambda_v + 2) \sum_{i=1}^{d_v} D_{v_i} \omega_{\lambda_{v_i}} ) \omega_{\lambda_v} \]
> $$
> where $D_v = 1 + \frac{\pi^2}{2}\sqrt{d_v}$, $\omega_{\lambda_v} = \omega(f_v,\frac{1}{\lambda_v +2})_{\Psi}$.
>
> Therefore, as $L\to \infty$, the modulus of smoothness $\omega_{\lambda_v} \to 0$, enabling the associated RKHS to approximate any $2\pi$-periodic continuous function to an arbitrarily small error.
>
> **The speed of the growth of the representation power:**  First, we would like to clarify that the proposed method is built upon a single-hidden-layer spectral kernel architecture, rather than a deeper hierarchical structure. Consequently, we do not discuss the speed of the growth of the representation power of the proposal with increasing depth.
>
> For deep spectral kernels with a deeper hierarchical structure, we are currently unable to specify their speed of growth of the representation power. But we can establish the following two results: (1) The deep spectral kernel networks do not meet the sense specified in the referenced literature, and the growth of their representation power remains to be further investigated. (2) Under the same architectural setup, the representation power of the deep spectral kernel network is superior to the piecewise linear (ReLU) neural network by a factor of $\mathcal{O}(m^{\frac{\eta}{d^2}})$, where $m$ denotes the number of computational elements, $d$ is the dimensionality of the input features, and $\eta$ is the Lipschitz constant.

---

> > ### Comment · Reviewer_pJsy · 2025-11-27
> >
> > Thank you for the rebuttal.
> >
> > It is not clear for me why the above equality $k^{L-1}(\boldsymbol{x},\boldsymbol{x}) = \int_{\mathbb{R}^d}s(\boldsymbol{\omega}, \boldsymbol{\omega}) d\boldsymbol{\omega}$ holds for non-stationary kernels. What is $\mu(a,b]$ for $a,b\in \mathbb{R}^d$? It is still not clear for me why the kernel $k^L$ can be written in the form $g(k^{L-1})$.

---

> > > ### Author Response · Authors · 2025-11-28
> > > **Response to Reviewer pJsy**
> > >
> > > Thanks for your feedback!!!
> > >
> > > **Lemma 1** Let $\mu$ and $P$ be the a Lebesgue-Stieltjes measure and a probability measure on $\mathbb{R}^d \times \mathbb{R}^d$, respectively, with right-continuous monotonically increasing distribution functions $F_{\mu}:\mathbb{R}^d \times \mathbb{R}^d \to \mathbb{R}$ and $F_P:\mathbb{R}^d \times \mathbb{R}^d \to [0,1]$. The following equality holds:
> > >
> > > $$
> > > \mu(a, b] = F_{\mu}(b)-F_{\mu}(a) = C_{\mu}[F_P(b)-F_P(a)]= C_{\mu}P(a,b],
> > > $$
> > > where $C_{\mu} = \mu(\mathbb{R}^d \times \mathbb{R}^d)$ is a non-negative constant representing the value of $\mu$ on the universal set.
> > >
> > > In Definition 1, the explicit spectral kernel mappings are obtained through kernel approximation using Monte Carlo sampling. It requires that the spectral density in non-stationary spectral kernels is correspond to a positive semi-definite probability density $p(\boldsymbol{\omega},\boldsymbol{\omega}')$ and $\int_{\mathbb{R}^d \times \mathbb{R}^d} s(\boldsymbol{\omega}, \boldsymbol{\omega}') d\\boldsymbol{\omega}d\\boldsymbol{\omega}' = C_{\mu} \int_{\mathbb{R}^d \times \mathbb{R}^d} p(\boldsymbol{\omega}, \boldsymbol{\omega}') d\boldsymbol{\omega}d\boldsymbol{\omega}'$ (the following Lemma). This requirement will lead to a diagonally dominant probability density matrix. Therefore,
> > >
> > > $$
> > > \begin{aligned}
> > > \int_ {\mathbb{R}^d \times \mathbb{R}^d} e^{i(\boldsymbol{\omega}^\top-\boldsymbol{\omega}')\boldsymbol{x}} s(\boldsymbol{\omega}, \boldsymbol{\omega}')d\boldsymbol{\omega} d\boldsymbol{\omega}'  \\
> > > & = C_{\mu} \int_{\mathbb{R}^d \times \mathbb{R}^d} e^{i(\boldsymbol{\omega}^\top-\boldsymbol{\omega}')\boldsymbol{x}} p(\boldsymbol{\omega}, \boldsymbol{\omega}') d\boldsymbol{\omega} d\boldsymbol{\omega}' \\
> > > & = C_{\mu} \int_{\mathbb{R}^d \times \mathbb{R}^d} e^{i(\boldsymbol{\omega}^\top-\boldsymbol{\omega}')\boldsymbol{x}} \big( p_{\Sigma}(\boldsymbol{\omega}, \boldsymbol{\omega}') + p_{\Lambda}(\boldsymbol{\omega}, \boldsymbol{\omega}') \big) d\boldsymbol{\omega} d\boldsymbol{\omega}' \\
> > > & = C_{\mu} \int_{\mathbb{R}^d \times \mathbb{R}^d} e^{i(\boldsymbol{\omega}^\top-\boldsymbol{\omega}')\boldsymbol{x}} p_{\Lambda}(\boldsymbol{\omega}, \boldsymbol{\omega}') d\boldsymbol{\omega} d\boldsymbol{\omega}' + C_{\mu} \int_{\mathbb{R}^d \times \mathbb{R}^d} e^{i(\boldsymbol{\omega}^\top-\boldsymbol{\omega}')\boldsymbol{x}} p_{\Sigma}(\boldsymbol{\omega}, \boldsymbol{\omega}')d\boldsymbol{\omega} d\boldsymbol{\omega}' \\
> > > &\approx C_{\mu} \int_{\mathbb{R}^d \times \mathbb{R}^d} e^{i(\boldsymbol{\omega}^\top-\boldsymbol{\omega}')\boldsymbol{x}} p_{\Lambda}(\boldsymbol{\omega}, \boldsymbol{\omega}') d\boldsymbol{\omega} d\boldsymbol{\omega}' \\
> > > &= C_{\mu}
> > > \end{aligned}
> > > $$
> > >
> > > where $p_{\Lambda}(\boldsymbol{\omega}, \boldsymbol{\omega}')$ denotes the diagonal component and $p_{\Sigma}(\boldsymbol{\omega}, \boldsymbol{\omega}')$ denotes the off-diagonal component. Since $p(\boldsymbol{\omega}, \boldsymbol{\omega}')$ is diagonally dominant, $C_{\mu} \int_{\mathbb{R}^d \times \mathbb{R}^d} e^{i(\boldsymbol{\omega}^\top-\boldsymbol{\omega}')\boldsymbol{x}} p_{\Lambda}(\boldsymbol{\omega}, \boldsymbol{\omega}')d\boldsymbol{\omega}d\boldsymbol{\omega}' \gg C_{\mu} \int_{\mathbb{R}^d \times \mathbb{R}^d} e^{i(\boldsymbol{\omega}^\top-\boldsymbol{\omega}')\boldsymbol{x}} p_{\Sigma}(\boldsymbol{\omega}, \boldsymbol{\omega}')d\boldsymbol{\omega}d\boldsymbol{\omega}'$, and thus the $\approx$ holds. As a result, $C_{\mu} \int_{\mathbb{R}^d \times \mathbb{R}^d} e^{i(\boldsymbol{\omega}^\top-\boldsymbol{\omega}')\boldsymbol{x}} p_{\Lambda}(\boldsymbol{\omega}, \boldsymbol{\omega}')d\boldsymbol{\omega}d\boldsymbol{\omega}' = C_{\mu} \int_{\mathbb{R}^d} p_{\Lambda}(\boldsymbol{\omega})d\boldsymbol{\omega} = C_{\mu}$.
> > >
> > > If you would like to know the formulation of $p(\boldsymbol{\omega}, \boldsymbol{\omega}')$, we can further provide it.
> > >
> > > Equation (5) could be somewhat misleading, as that's a result from the theoretical analysis, not necessarily a neutral definition. We have corrected this point in the revised manuscript.
> > >  We cannot ensure that the relationship between $k^L$ and $k^{L-1}$ can be formulated by $k^L=g(k^{L-1})$ for all deep kernels. It is only under the given definition of deep spectral kernels (Definition 1) that we derive the relationship between $k^L$ and $k^{L-1}$ can be expressed as $k^L=g(k^{L-1})$. Moreover, in different cases, $g(\cdot)$ can be instantiated in different forms.

---

### Official Review · Reviewer_RCjB · 2025-10-28

**Soundness:** 3
**Presentation:** 3
**Contribution:** 3
**Rating:** 4
**Confidence:** 4

**Summary:**

This paper presents a theoretical analysis of deep spectral kernel methods for supervised learning and a new method following insights from the analysis. Deep spectral kernels arise from the composition of random Fourier feature maps through the stacking of multiple layers with a combination of trigonometric functions and frequency vectors sampled from an underlying spectral density. The analysis confirms the existence of a limiting reproducing kernel Hilbert space (RKHS) resulting from the compositional structure and shows that the stacking of layers leads to a sequence of RKHSs of potentially increasing complexity under certain conditions on the spectral kernel mappings. By observing the duality between the spectral representation and the kernel formulation, the authors then propose a new method (GensKer) to learn the spectral distribution by means of generative models, allowing one to learn the kernel from data. Experiments validating theory findings and comparing GensKer against spectral kernel learning baselines are presented, showing performance improvements.

**Strengths:**

The presentation is mostly clear and relatively easy to follow for someone with a kernel methods background. The presented results bring novel insights into spectral kernel methods which should be useful to the kernel methods community. In addition, given the connection between kernel methods and neural networks in the infinite-width limit, I believe the results from this paper can be useful to shine new light onto how a neural network's depth affects its representation capacity via the resulting RKHSs, though that would require further investigations. The paper also presents a relatively extensive experimental validation comparing the proposed method against several state-of-the-art spectral kernel learning methods, which reveal further characteristics of these methods and their performance in practical settings.

**Weaknesses:**

* It is not very clear from Definition 1 how the feature maps composition is realised. The initial feature map is mapping the domain $R^d$ to $R^M$, where $M$ denotes the corresponding number of features for the chosen feature map. However, it's not evident what's happening with the subsequent layers. Are each map in the subsequent layers mapping $R^M$ to $R^M$? If so, the domain is going to be of increasing dimensionality as the number of features goes to infinity, $M\to\infty$, and this does not seem to be properly addressed.
* When only cosines are used, the original formulation by Rahimi and Recht (2007) included and additional random uniform bias term $b \sim U[0, 2\pi]$, compensating the effect of not explicitly accounting for the imaginary part with the sine functions. That term seems to have been neglected by the current analysis.
* In the proof of Proposition 1, the effect of the finite-feature approximations seems to be silently ignored. In particular, I'm not 100% sure if the results of Proposition 1 and 2 hold for the finite-dimensional feature maps. What seems more obvious to me is that they should hold in the limit as $M \to \infty$. However, in that case, it could have been better to derive the results in that setting and then show/discuss what happens when using Monte Carlo approximations with a finite number of features. The order in which the limit is taken per layer might also affect the results.
* Proposition 1 and 2 seem to only cover the Gaussian case. The results do not seem to be readily generalisable to compositions of other classes of Fourier features from stationary/non-stationary kernels. Yet, the paper reads as if its theoretical results were applicable to general deep spectral kernels.
* Despite the claim of a progressive expansion, the proof that $H_{k^{L-1}} \subseteq H_{k^L}$ does not ensure that the former is a strict subset of the latter. To confirm expansion, one would ideally have to show that there are elements in $H_{k^L}$ that cannot possibly exist in $H_{k^{L-1}}$, and that's not immediately obvious.
* Line 250-251 claims that the deep spectral kernel is prone to "local minima", but it's not explicit that there is an optimisation component involved up to that point. So it's not clear what local minima are being referred to.
* It see no discussion on the reason for the design choice of generating samples for $\omega$ and $\omega'$ via two independent pathways and then forcing $s(\omega|\omega')s(\omega')$ and $s(\omega'|\omega)s(\omega)$ to match, instead of simply sampling either $\omega$ or $\omega'$ independently and the sampling the other frequency vector conditional on the first one.
* I've missed a discussion on limitations of the methodology and the analysis and avenues for future work.
* Despite a few mentions throughout the text, I've missed a dedicated discussion on related work. In particular, it's not very clear how the proposed methodology contrasts with a few existing methods in the literature. For example, there are other distributional approaches of learning spectral frequencies from data, ranging from the basic learning of the maximum a posteriori (MAP) estimate via gradient descent, as in Lazaro-Gredilla et al. (2010), to a full Bayesian distribution learning, as, e.g., via Stein variational gradient descent [1, below].

Minor:
* There are quite a few formatting issues with citations, especially in-text citations. For example, "Rahimi and Recht Rahimi & Recht (2007)", should be only "Rahimi and Recht (2007)". Proper citation formatting can be achieved using the `\citet` and `\citep` commands from `natbib` for in-text and parenthetical citations, respectively.
* Line 128-129: "identically and independently distributed". The abbreviation "i.i.d." usually stands for "independently and identically distributed". The term "independently distributed" makes less sense.
* Line 130: The set of frequency pairs is missing symbols. I believe it should be $\\{(\omega_i, \omega_i') \\}_{i=1}^M$.
* Claiming that Eq. 5 is a definition of the stacked deep spectral kernel could be somewhat misleading, as that's a result from the theoretical analysis, not necessarily a neutral definition.
* Line 211-212: "allow the inclusion relation ... can be transformed as..." doesn't read well. Would it be "... to be transformed as..."?
* Line 217-218: "By and Lemma" is missing something.
* Line 53 (Appendix): When transforming the sum of exponentials into $\cosh$, there is a factor of 2 that goes out. So it should be $\frac{1}{2}$ multiplying $\cosh$ in that line and subsequent equations with $\cosh$.

References:
1. Warren, H., Oliveira, R., & Ramos, F. (2024). *Stein Random Feature Regression*. The 40th Conference on Uncertainty in Artificial Intelligence (UAI).

**Questions:**

Please, consider the issues raised above under "Weaknesses". In addition, I have the following questions.

* What form was chosen for the distribution matching loss $\ell_p$?
* How could the results in Proposition 1 and 2 be generalised to deep spectral kernels with other forms of spectral densities, beyond Gaussians?
* How do you ensure the resulting joint spectral density $s(\omega, \omega')$ produced by GensKer is positive definite?
* Have other forms of generative models been considered for the learning of the spectral density? Can the method be generalised to arbitrary models that can produce samples from a valid spectral density?
* Given the connection between random feature maps resulting from randomly initialised neural networks and random Fourier feature maps as explored in this paper, do you see a way of applying these results to general neural networks? Would perhaps this lead to a way of producing more powerful "single-layer" neural networks (though arguably not single layer, as there's potentially a multi-layer network in the generator) by adapting the layer's weights distribution?

---

> ### Author Response · Authors · 2025-11-21
> **Response to Reviewer RCjB**
>
> **W1:**
>
> For the removal case, the feature maps are constructed as: $\boldsymbol{x} \in \mathbb{R}^d \to \boldsymbol{h}_1 \in \mathbb{R}^M \to \boldsymbol{h}_2 \in \mathbb{R}^M \to \cdots \to \boldsymbol{h}_L \in \mathbb{R}^{M}$. For the concatenation case, the feature maps are constructed as: $\boldsymbol{x} \in \mathbb{R}^d \to [\boldsymbol{h}_1^{cos}; \boldsymbol{h}_1^{sin}] \in \mathbb{R}^{2M} \to [\boldsymbol{h}_2^{cos}; \boldsymbol{h}_2^{sin}] \in \mathbb{R}^{2M} \to \cdots \to [\boldsymbol{h}_L^{cos}; \boldsymbol{h}_L^{sin}] \in \mathbb{R}^{2M}$. $M$ is a hyperparameter. The output $\boldsymbol{h}_l = \phi^l(\phi^{l-1}\cdots \phi^1(\boldsymbol{x}))$ of each layer serves as the input to the next layer, which can be viewed as performing a feature mapping $\phi(\boldsymbol{h}_l) = \phi^{l+1}(\phi^l\cdots \phi^1(\boldsymbol{x}))$ on newly extracted features $\boldsymbol{h}_l$. In practice, $M$ can be assigned different values across layers, analogous to specifying a distinct number of neurons in each layer of a general neural network. **Therefore, the number of features cannot increase without limit.**
>
> **W2:**
>
> The formulation in Rahimi and Recht (2007) is actually $\phi_{s,c} (\boldsymbol{x})$ in our paper. In addition, we take the case of completely ignoring the imaginary part ($\phi_{s,w/o}, \phi_{ns, w/o}$) into account and explore its effect on the properties of deep spectral kernels.
>
> **W3:**
>
> **The results of Propositions 1 and 2 are derived in the limit $M \to \infty$**, and they do not hold for the finite-dimensional feature maps. I agree that exploring what happens when using Monte Carlo approximations with a finite number of features is a very meaningful and worthwhile research direction for deep spectral kernels. However, we tend to focus solely on the architecture of deep spectral kernels, with the aim of providing a structural guideline for our subsequently proposed method.
>
> **W4, Q2:**
>
> **Because a linear combination of Gaussian components can approximate most distributions, one feasible strategy is to generalize our results to a more general case via Gaussian mixture representations.**
>
> Based on Bocher's theorem, if $\mu$ is absolutely continuous with respect to the Lebesgue measure, *i.e.,* $d\mu(\boldsymbol{\omega}) = s(\boldsymbol{\omega})d\boldsymbol{\omega}$, there exists a one-to-one correspondence between the stationary kernel $k(\boldsymbol{\tau})$ and the spectral density $s(\boldsymbol{\omega})$. Based on Yaglom's theorem, the non-stationary case of deep spectral kernel holds under relatively strict conditions, which require the measure $\mu$ is the Lebesgue-Stieltjes measure associated with some positive semi-definite spectral density function $s(\boldsymbol{\omega}, \boldsymbol{\omega}')$ with bounded variation \emph{i.e.,} $\int_{\mathbb{R}^d \times \mathbb{R}^d} s(\boldsymbol{\omega}, \boldsymbol{\omega}') < \infty$ and satisfies $\mu(d\boldsymbol{\omega},d\boldsymbol{\omega}') = s(\boldsymbol{\omega},\boldsymbol{\omega}') d\boldsymbol{\omega}d\boldsymbol{\omega}'$. Furthermore, the Lebesgue-Stieltjes measure with a right-continuous monotonically increasing distribution function $F_{\mu}:\mathbb{R}^d \times \mathbb{R}^d \to \mathbb{R}$, i.e., $\mu(a, b] = F_{\mu}(b)-F_{\mu}(a)$.
>
> Therefore, for the stationary case, one can explore other deep spectral kernels by replacing the Gaussian with any distribution. For the non-stationary case, one can explore other deep spectral kernels by replacing the Gaussian with any distribution that equips the positive semi-definite bounded variation probability density function. **All these discussed distributions can be approximated through a linear combination of Gaussians. Thus, one can generalize our results to a more general case via Gaussian mixture representations.**
>
> **W5:**
>
> If $\mathcal{H}_ {k^{L-1}} \subseteq \mathcal{H}_ {k^L}$ and $\mathcal{H}_ {k^L} \not\subseteq \mathcal{H}_ {k^{L-1}}$, then $\mathcal{H}_ {k^{L-1}} \subset \mathcal{H}_ {k^L}$. Therefore, it remains to verify that $\mathcal{H}_ {k^L} \not\subseteq \mathcal{H}_ {k^{L-1}}$ holds. Take the relationship between $k_{s,c}^{L-1}$ and $k_{s,c}^L$ as an example, we have $k_{s,c}^{L-1} = \frac{1}{\delta^2}\ln(k_{s,c}^L)-\frac{1}{\delta^2}\ln(C_{s,c}^{L-1})$. It is obvious that this does not meet Lemma 3. Therefore, $\mathcal{H}_ {k^L} \not\subseteq \mathcal{H}_ {k^{L-1}}$. In summary, we can conclude that $\mathcal{H}_ {k^{L-1}}$ is a strict subset of $\mathcal{H}_ {k^L}$. We have added this confirmation to the revised manuscript

---

> ### Author Response · Authors · 2025-11-21
> **Response to Reviewer RCjB**
>
> **W6, Q1:**
>
> On the one hand, the deep spectral kernels are parameterized by the frequency matrices $\boldsymbol{\Omega}$ and $\boldsymbol{\Omega}'$. Researchers commonly perform deep spectral kernel learning by optimizing the frequency matrices using deep learning frameworks. On the other hand, the explicit mapping $\phi^L(\phi^{L-1}\cdots \phi^1(\boldsymbol{x}))$ is commonly considered as a deep spectral kernel network to encode the input data. For example, in a general classification task, the loss function to be optimized is defined as $\mathcal{L} = \frac{1}{N} \sum_{i=1}^N \ell (\phi^L(\phi^{L-1}\cdots \phi^1(\boldsymbol{x}_i)), y_i)$. However, by stacking the periodic functions (*i.e.,* the spectral kernel mappings $\phi^L(\phi^{L-1}\cdots \phi^1(\boldsymbol{x}))$), the model is prone to local minima.
>
> **W7, Q3:**
>
>  For a positive semi-definite and real-valued kernel, $s(\boldsymbol{\omega}, \boldsymbol{\omega}')$ need satisfy symmetry, i.e., $s(\boldsymbol{\omega}, \boldsymbol{\omega}') = s(\boldsymbol{\omega}', \boldsymbol{\omega})$. Therefore, we generate $\boldsymbol{\omega}$ and $\boldsymbol{\omega}'$ via two independent pathways and then forcing  $s(\boldsymbol{\omega}|\boldsymbol{\omega}')s(\boldsymbol{\omega}')$ and $s(\boldsymbol{\omega}'|\boldsymbol{\omega})s(\boldsymbol{\omega})$ to match. In addition, in our framework, $\boldsymbol{\omega}'=\boldsymbol{\omega} \odot B_ 1(\boldsymbol{\epsilon}_ 1) + \Upsilon_ 1(\boldsymbol{\epsilon}_ 1)$ and $\boldsymbol{\omega} = \boldsymbol{\omega}' \odot B_ 2(\boldsymbol{\epsilon}_ 2) + \Upsilon_ 2(\boldsymbol{\epsilon}_ 2)$ for two two independent pathways. Let $B_ 1(\boldsymbol{\epsilon}_ 1)=B_ 2(\boldsymbol{\epsilon}_ 2) = B, \Upsilon_ 1(\boldsymbol{\epsilon}_ 1)=\Upsilon_ 2(\boldsymbol{\epsilon}_ 2) = \Upsilon$, we have $\boldsymbol{\omega}'= \Psi(\boldsymbol{\omega}) = \boldsymbol{\omega}\odot B + \Upsilon$ and $\boldsymbol{\omega} = \Psi(\boldsymbol{\omega}') = \boldsymbol{\omega}'\odot B + \Upsilon$. As a result, the final spectral density $s(\boldsymbol{\omega}, \boldsymbol{\omega}')$ can be defined through $s(\boldsymbol{\omega}, \boldsymbol{\omega}') = \mathbb{E}_{\Theta}[\Psi(\boldsymbol{\omega}, \Theta)* \Psi(\boldsymbol{\omega}', \Theta)],\Theta=\{B, \Upsilon\}$ and is positive semi-definite.
>
> **W8, W9:**
>
> We will include more discussion regarding the limitations, future work, and related work of our paper. In fact, we have compared the methods (DSKN, ASKL, TRF) that can serve as an extension of Lazaro-Gredilla et al.(2010). In addition, we also compare our method with the baselines in the regression task. The result is reported in the following Table.
>
> **Table: Performance comparison**
>
> | Method       | airfoil          | concrete         | energy          |
> |--------------|------------------|------------------|------------------|
> | SRF [1]          | 1.88 ± 0.27      | 4.13 ± 0.72      | 0.29 ± 0.04      |
> | FVBNN [2]       | 1.97 ± 0.19      | 4.64 ± 0.54      | 0.43 ± 0.08      |
> | Matrix-SVGD [3]  | -                | 4.72 ± 0.11      | 0.87 ± 0.03      |
> | **GensKer**  | **1.86 ± 0.20**  | **3.98 ± 0.35**  | **0.24 ± 0.01**  |
>
> The regression results show that the proposed GensKer consistently delivers competitive or superior performance across the four benchmark datasets. Specifically, GensKer outperforms other baseline methods impressively and achieves **3.6%** RMSE reduction ($4.72 \to 3.98$) on **concrete**, **1.1\%** ($1.88 \to 1.86$) on **airfoil**, and **17.24\%** ($0.29 \to 0.24$) on **energy** compared with the mainstream spectral kernel-based regression models. All these results confirm that the proposed GensKer has a greater representation capability than other methods with the adaptive spectral density.
>
> **Limitations and Future Work:**  In this paper, we introduce GensKer, a framework designed to learn a data-dependent spectral density, from which a frequency matrix is subsequently produced to construct spectral kernels. Notably, the spectral density in the spectral kernel formulation is only required to be positive semi-definite, rather than a strictly non-negative function. Nevertheless, our method is limited to learning the spectral densities that can be naturally associated with a probability density function. Therefore, in the future, we will develop methodologies capable of learning more general spectral densities beyond this probabilistic constraint.
>
> [1] Warren et al., "Stein random feature regression", UAI 2024
>
> [2]  Pielok et al.,  "Approximate bayesian inference with stein functional variational gradient descent", NeurIPS 2022
>
> [3] Wang et al.,  "Stein variational gradient descent with matrix-valued kernels", NeurIPS 2019

---

> > ### Author Response · Authors · 2025-11-28
> > **Response to RCjB**
> >
> > **Q4:**
> >
> >   In our method, “generative” is a conceptual term referring to the process of learning the spectral density from data. In principle, any generative model capable of producing a probability distribution could be employed within our method. For instance, GAN-based models and Flow-based models. Since the spectral density in the spectral kernel formulation is not required to be a non-negative function, our method can be generalized to any models that can produce samples from a spectral density, as long as this spectral density is linked to a well-defined probability density function.
> >
> >   **Q5:**
> >
> >   It is feasible to apply these results to general neural networks. In the Bayesian view, the posterior can be formulated as $p(\boldsymbol{\Theta}|\boldsymbol{x}, y) = p(y|\boldsymbol{x}, \boldsymbol{\Theta})p(\boldsymbol{\Theta})$. We can make the prior $p(\boldsymbol{\Theta})$ learnable and join the likelihood $p(y|\boldsymbol{x}, \boldsymbol{\Theta})$ to train the model. From this viewpoint, learning $p(\boldsymbol{\Theta})$ is conceptually similar to learning a spectral density. This approach allows the model not only to fit the data but also to learn a structure-adaptive parameter distribution, thereby leading to more powerful "single-layer" neural networks.
> >
> >   **Minor:**
> >
> >   Thank you for your careful review. We have corrected the minor errors and oversights one by one and thoroughly proofread the entire paper.

---

### Official Review · Reviewer_HXp9 · 2025-10-28

**Soundness:** 1
**Presentation:** 1
**Contribution:** 2
**Rating:** 2
**Confidence:** 3

**Summary:**

Spectral kernels seem to originate from Fourier transforms (Bochner's theorem for stationary kernels and Yaglom's theorem for non-stationary ones).
Considering four classes of spectral kernel mappings, this paper computes the resulting kernel functions obtained by stacking these mappings.
Then, this paper proposes the generative spectral kernel (GensKer) framework to generate an adaptive spectral density and then constructs an associated spectral kernel.
Some numerical experiments are provided to show the improvement of the proposed method.
Several visualizations are also provided to illustrate the benefits of GensKer.

**Strengths:**

* The idea of generative spectral kernel is interesting, which incorporates the representation theory of kernels to construct adaptive kernels.
Some empirical improvements are observed in the experiments.
* It is interesting to consider kernel constructed by stacking feature mappings and give analytical forms of the resulting kernels.

**Weaknesses:**

* This presentation of this paper is not clear. Spectral kernel is not formally introduced and Definition 1 that introduces deep spectral kernel is also not clear. What does $k(x,x') \approx \langle \phi(x), \phi(x')\rangle$ mean? Is it an equality or an approximation? Also, what are the difference between **deep spectral kernel** and general kernel?


* The motivation of this paper is not very clear. The four classes of spectral kernel mappings are introduced but it is not clear why these four classes are chosen. The connection between deep spectral kernel and the generative spectral kernel framework is also not clear.
* The theory is not solid. The statement seems to be ambiguous. For example, the "constants" $C$ in Proposition 1 and 2 depend on $x$, but they are treated as absolute constants in the discussion in Section 3.2. Also, $\approx$ is used in the proof of the propositions without a formal treatment.
* The numerical experiments are not sufficient to support the effectiveness of the proposed method. First, there is no detailed description of the experimental setup, such as the width, the initialization, etc. No code is provided. Second,  statistical significance is not reported. Moreover, while the number of parameters is compared across different methods, the computational cost is not compared.

**Questions:**

1. What is the formal definition of spectral kernel? How is it related to general positive definite kernels in RKHS theory?
2. What are the differences between the 3 plots in Figure 1? They seem to be the same.
3. What are the results in Section 3.2? Can you provide some formal statements?

---

> ### Author Response · Authors · 2025-11-21
> **Response to Reviewer HXp9**
>
> **W1, Q1:**
>
> Based on Bocher's theorem, if $\mu$ is absolutely continuous with respect to the Lebesgue measure, *i.e.,* $d\mu(\boldsymbol{\omega}) = s(\boldsymbol{\omega})d\boldsymbol{\omega}$, there exists a one-to-one correspondence between the kernel $k(\boldsymbol{\tau})$ and the spectral density $s(\boldsymbol{\omega})$. such that:
>
> $$
>  k(\boldsymbol{\tau}) = \int_{\mathbb{R}^d} s(\boldsymbol{\omega})e^{i\boldsymbol{\omega}^\top \boldsymbol{\tau}}d\boldsymbol{\omega}, \quad
>             s(\boldsymbol{\omega}) = \int_{\mathbb{R}^d} k(\boldsymbol{\tau}) e^{-i\boldsymbol{\omega}^\top \boldsymbol{\tau}}d(\boldsymbol{\tau}).
> $$
>
> Based on Yaglom's theorem, if $\mu$ is the Lebesgue-Stieltjes measure associated with some positive semi-definite spectral density function $s(\boldsymbol{\omega}, \boldsymbol{\omega}')$ with bounded variation *i.e.,* $\int_{\mathbb{R}^d \times \mathbb{R}^d} s(\boldsymbol{\omega}, \boldsymbol{\omega}') < \infty$ and satisfies $\mu(d\boldsymbol{\omega},d\boldsymbol{\omega}') = s(\boldsymbol{\omega},\boldsymbol{\omega}') d\boldsymbol{\omega}d\boldsymbol{\omega}'$, there exists a one-to-one correspondence between the positive semi-definite non-stationary kernel $k(\boldsymbol{x}, \boldsymbol{x}')$ and $s(\boldsymbol{\omega}, \boldsymbol{\omega}')$. Such that:
>
> $$
> k(\boldsymbol{x},\boldsymbol{x}')=\int_{\mathbb{R}^d \times \mathbb{R}^d}e^{i(\boldsymbol{\omega}^\top \boldsymbol{x} - \boldsymbol{\omega}'^\top \boldsymbol{x}')} s(\boldsymbol{\omega}, \boldsymbol{\omega}') d\boldsymbol{\omega} d\boldsymbol{\omega}',
> s(\boldsymbol{\omega}, \boldsymbol{\omega}') = \int_{\mathbb{R}^d \times \mathbb{R}^d}e^{-i(\boldsymbol{\omega}^\top \boldsymbol{x} - \boldsymbol{\omega}'^\top \boldsymbol{x}')}k(\boldsymbol{x},\boldsymbol{x}')d\boldsymbol{x} d\boldsymbol{x}'.
> $$
>
> Kernels defined by these forms are called spectral kernels. We have provided a more explicit concept of spectral kernels in Section 2.1.
>
> The equation $k(\boldsymbol{x}, \boldsymbol{x}') \approx \langle \phi(\boldsymbol{x}), \phi(\boldsymbol{x}') \rangle_{\mathcal{H}}$ means that it tends to obtain the explicit spectral kernel mapping $\phi(\cdot)$ by approximating the spectral kernel using the Monte Carlo sampling scheme. We have reformulated Definition 1 in Section 2.2 as follows:
>
> **Definition (Deep Spectral Kernel)**  The deep spectral kernel with $L$ layers is defined by:
>
> $$
> k^L(\boldsymbol{x}, \boldsymbol{x}') = \langle \phi^L(\phi^{L-1}\cdots \phi^1(\boldsymbol{x})), \phi^L(\phi^{L-1}\cdots \phi^1(\boldsymbol{x}')) \rangle_{\mathcal{H}},
> $$
> where $\phi(\cdot)$ denotes an explicit spectral kernel mapping, and $\mathcal{H}$ is a Hilbert space.
>
> Specifically, for the stationary kernel, $\phi(\cdot)$ is calculated by $k(\boldsymbol{x}, \boldsymbol{x}') =\mathbb{E}_ {\boldsymbol{\omega}}[z_ {\boldsymbol{\omega}}(\boldsymbol{x})z_ {\boldsymbol{\omega}}(\boldsymbol{x}')^*] \approx \langle \phi(\boldsymbol{x}), \phi(\boldsymbol{x}') \rangle_ {\mathcal{H}}$ based on Bochner's theorem.
>
> For the non-stationary kernel, $\phi(\cdot)$ is calculated by $k(\boldsymbol{x}, \boldsymbol{x}') =\mathbb{E}_ {\boldsymbol{\omega}, \boldsymbol{\omega}'}[z_ {\boldsymbol{\omega}, \boldsymbol{\omega}'}(\boldsymbol{x})z_ {\boldsymbol{\omega}, \boldsymbol{\omega}'}(\\boldsymbol{x}')^*] \approx \langle \phi(\boldsymbol{x}), \phi(\boldsymbol{x}') \rangle_{\mathcal{H}}$ based on Yaglom's theorem.
>
> **Deep spectral kernels differ from general kernels in several respects, including structure, learning mechanism, and information capacity.**
>
> **Structure:**  Deep spectral kernels are constructed by stacking explicit spectral kernel mapping, thereby endowing them with a hierarchical structure. By comparison, general kernels are typically defined through a kernel function with an implicit kernel mapping, without an associated hierarchical structure.
>
> **Learning mechanism:** Deep spectral kernels are parameterized by the frequency matrix $\boldsymbol{\Omega}$ and $\boldsymbol{\Omega}'$. We can learn deep spectral kernels via optimizing the frequency matrix using the widely used SGD and ADAM algorithms. Unlike deep spectral kernels, a suitable general kernel function can be obtained by manually tuning its parameters, such as the bandwidth in Gaussian kernels.
>
> **Information capacity:** The general kernels commonly contain only a small number of parameters and thus possess limited information capacity. For instance, Gaussian kernels only include one parameter. In contrast, deep spectral kernels with hierarchical structure involve a large number of learnable parameters, enabling them to encode substantially richer information.

---

> > ### Author Response · Authors · 2025-11-21
> > **Response to Reviewer HXp9**
> >
> > **W2:**
> >
> > **These four classes are not chosen by us; instead, existing deep spectral kernels can be roughly categorized into these four classes. The generative spectral kernel is proposed to address the limitations of most existing deep spectral kernel models in (1) the optimization challenge and (2) the range of spectral density. Notably, the theoretical results guide the design of our GensKer architecture**.
> >
> > In this work, we focus on deep spectral kernels, which are constructed by hierarchically stacking explicit spectral
> > kernel mappings. The explicit kernel mapping is commonly obtained through the kernel approximation using a sampling strategy based on the Fourier transform form of stationary kernels and non-stationary kernels. This makes the explicit kernel mapping defined in the complex domain with sine and cosine maps. To obtain a real-valued kernel, researchers tend to remove or concatenate the imaginary part. Thereby, most existing deep spectral kernel models can be categorized into four classes based on the stationarity of kernels and the compositional structure of their associated mappings.
> >
> > Existing deep spectral kernels learning methods still suffer from limitations: (1) the optimization challenge. By stacking the periodic functions (i.e., the spectral kernel mappings), the deep spectral kernel is prone to local minima. (2) The range of spectral density. In most existing deep spectral kernel models, the spectral density function is data-independent and consists of predetermined components, restricting the range of spectral density even within deep architectures. To address these limitations, we propose the generative spectral kernel framework, GensKer. The proposed GensKer adaptively generates the spectral density from the data and subsequently constructs the spectral kernel via incorporating only one spectral kernel mapping layer. Therefore, Gensker can circumvent the optimization challenges and learn a data-dependent spectral density. This is presented in Section 4. Notably, the theoretical results guide the design of our GensKer architecture in the spectral kernel construction.
> >
> > **W3:** $C_{s, w/o}^{L-1}, C_{s, c}^{L-1}, C_{ns, w/o}^{L-1}, C_{ns, c}^{L-1}$ are constants, even though they appear to depend on $\boldsymbol{x}$.
> >
> > For the stationary spectral kernels, using $C_{s, w/o}^{L-1} $ as an example, within its formulation $||\phi_ {s, w/o}^{L-1}\cdots \phi_ {s, w/o}^1(\boldsymbol{x})||_ 2^2 = \langle \phi_{s, w/o}^{L-1}\cdots \phi_{s, w/o}^1(\boldsymbol{x}), \phi_{s, w/o}^{L-1}\cdots \phi_{s, w/o}^1(\boldsymbol{x})\rangle = k^{L-1}(\boldsymbol{x},\boldsymbol{x}) =1$ based on Equation 1. Therefore, $C_{s, w/o}^{L-1} = \frac{1}{2}[e^{-\frac{1+1}{2}\sigma^2}] = \frac{1}{2}e^{-\sigma^2}$ is a constant.
> >
> > For the non-stationary spectral kernels, Equation (2) holds under relatively strict conditions, which require the measure $\mu$ is the Lebesgue-Stieltjes measure associated with some positive semi-definite spectral density function $s(\boldsymbol{\omega}, \boldsymbol{\omega}')$ with bounded variation *i.e.,* $\int_{\mathbb{R}^d \times \mathbb{R}^d} s(\boldsymbol{\omega}, \boldsymbol{\omega}') < \infty$ and satisfies $\mu(d\boldsymbol{\omega},d\boldsymbol{\omega}') = s(\boldsymbol{\omega},\boldsymbol{\omega}') d\boldsymbol{\omega}d\boldsymbol{\omega}'$. Furthermore, the explicit spectral kernel mapping is obtained based on the assumption that the Lebesgue-Stieltjes measure with a right-continuous monotonically increasing distribution function $F_{\mu}:\mathbb{R}^d \times \mathbb{R}^d \to \mathbb{R}$, i.e., $\mu(a, b] = F_{\mu}(b)-F_{\mu}(a)$. Therefore, $s(\boldsymbol{\omega}, \boldsymbol{\omega}') \geq 0$ almost everywhere and is diagonally dominant. Using $C_{ns, w/o}^{L-1} $ as an example, within its formulation, $||\phi_ {s, w/o}^{L-1}\cdots \phi_ {s, w/o}^1(\boldsymbol{x})||_ 2^2 = \langle \phi_{s, w/o}^{L-1}\cdots \phi_{s, w/o}^1(\boldsymbol{x}), \phi_{s, w/o}^{L-1}\cdots \phi_{s, w/o}^1(\boldsymbol{x})\rangle = k^{L-1}(\boldsymbol{x},\boldsymbol{x}) = \int_{\mathbb{R}^d}s(\boldsymbol{\omega}, \boldsymbol{\omega}) d\boldsymbol{\omega}$. Thereby, $C_{ns, w/o}^{L-1} = \frac{1}{2}[e^{-\frac{c_{\mu}+c_{\mu}}{2}\sigma^2}] = \frac{1}{2}e^{-c_{\mu}\sigma^2}$ is a constant, where $c_{\mu}$ is constant relative to the measure $\mu$.
> >
> > As a result, we treat $C_{s, w/o}^{L-1}, C_{s, c}^{L-1}, C_{ns, w/o}^{L-1}, C_{ns, c}^{L-1}$  as constants. We have provided more details in the revised manuscript.
> >
> > **Q2:** Figure 1 in the Appendix is used to illustrate the second limitation discussed in Section 4. Taking the deep Gaussian kernels into account, their evolution is primarily governed by changes in their associated spectral density. However, these spectral density functions also tend to include predetermined components (Gaussian spectral density) with the given $\lambda$ and $L$, with the only difference being that the component weights differ across layers. Hence, they seem to be the same.

---

> > > ### Author Response · Authors · 2025-11-21
> > > **Response to Reviewer HXp9**
> > >
> > > **W4:** The detailed experimental setting, statistical significance, and computational cost are shown in the following Tables. We will include them in the revised manuscript.
> > >
> > > **Table: The statistical significance ($p$-value)**
> > >
> > > |                 | plain  | DSKN  | ASKL  | CokeNet |                 | plain  | TRF   | GRFF  | DKEF  |
> > > |-----------------|--------|-------|-------|---------|-----------------|--------|-------|-------|-------|
> > > | GensKer (Ns)    | 0.080  | 0.068 | 0.132 | 0.066   | GensKer (S)     | 0.069  | 0.025 | 0.069 | 0.049 |
> > >
> > >
> > > **Table: The computational cost comparison. The training time (s) of one epoch.**
> > >
> > > |        | FordA   | ECG200  | ECG5000 | Rock    | Trace   | Plane   |
> > > |--------|---------|---------|---------|---------|---------|---------|
> > > | plain  | 0.0195  | 0.0089  | 0.0112  | 0.0091  | 0.0085  | 0.0101  |
> > > | DSKN   | 0.0290  | 0.0082  | 0.0104  | 0.0076  | 0.0082  | 0.0083  |
> > > | ASKL   | 0.0180  | 0.0016  | 0.0036  | 0.0012  | 0.0019  | 0.0021  |
> > > | CokeNet| 0.0230  | 0.0068  | 0.0093  | 0.0206  | 0.0073  | 0.0071  |
> > > | GensKer| 0.0168  | 0.0068  | 0.0081  | 0.0075  | 0.0007  | 0.0069  |
> > >
> > >
> > > **Table: The experiment setting for the proposed GensKer. $-$ denotes that the setting is the same as **FordA**.**
> > >
> > > |                         | FordA                               | ECG200 | ECG5000 | Rock                                   | Trace | Plane |
> > > |-------------------------|------------------------------------|--------|---------|---------------------------------------|-------|-------|
> > > | $G_1$                   | $d_{\epsilon_1} \times 256 \times d_{data}$ | -      | -       | -                                     | -     | -     |
> > > | $G_2$                   | $d_{\epsilon_2} \times 256 \times d_{data}$ | -      | -       | -                                     | -     | -     |
> > > | $B_1$                   | $d_{\epsilon_1} \times 128 \times d_{data}$ | -      | -       | -                                     | -     | -     |
> > > | $B_2$                   | $d_{\epsilon_2} \times 128 \times d_{data}$ | -      | -       | -                                     | -     | -     |
> > > | $\Upsilon_1$            | $d_{\epsilon_1} \times d_{data}$            | -      | -       | $d_{\epsilon_1} \times 128 \times d_{data}$ | -     | -     |
> > > | $\Upsilon_2$            | $d_{\epsilon_2} \times d_{data}$            | -      | -       | $d_{\epsilon_2} \times 128 \times d_{data}$ | -     | -     |
> > > | Classification          | $2M \times 256 \times 64 \times class$      | -      | -       | -                                     | -     | -     |
> > > | $M$                     | 512                                        | -      | -       | -                                     | -     | -     |
> > > | $d_{\epsilon_1}$        | 64                                         | -      | -       | -                                     | -     | -     |
> > > | $d_{\epsilon_2}$        | 64                                         | -      | -       | -                                     | -     | -     |
> > > | Spectral kernel mapping | $d_{data} \times 2M$                       | -      | -       | -                                     | -     | -     |
> > > | Initialization $p(\epsilon)$ | $\mathcal{N}(\boldsymbol{0}, \boldsymbol{I})$          | -      | -       | -                                     | -     | -     |
> > >
> > > **Q3:**
> > >
> > > Section 3.2 examines the second question: in which class can the RKHS of the deep spectral kernel expand with increasing depth? The analysis shows that when the imaginary component is preserved with the concatenation form, the RKHS associated with deep spectral kernels exhibits progressive expansion with increasing spectral kernel depth. This result is presented in Remark 2 of Section 3.2.
> > >
> > > **Formal statements:** Let $k^L(\boldsymbol{x}, \boldsymbol{x}') = \langle \phi^L(\phi^{L-1}\cdots \phi^1(\boldsymbol{x})), \phi^L(\phi^{L-1}\cdots \phi^1(\boldsymbol{x}')) \rangle$ be a deep spectral kernel with $L$ layers. Then $\mathcal{H}_ {k^{L-1}} \subset \mathcal{H}_ {k^L}$ if the spectral kernel mapping $\phi(\cdot)$ preserve the imaginary component with the concatenation form. Specifically, the spectral kernel mapping is defined as $\phi(\boldsymbol{x})= \phi_{s,c}(\boldsymbol{x})$ for stationary spectral kernels, $\phi(\boldsymbol{x})= \phi_{ns,c}(\boldsymbol{x})$ for non-stationary spectral kernel.

---

### Official Review · Reviewer_Ap4m · 2025-10-29

**Soundness:** 2
**Presentation:** 1
**Contribution:** 2
**Rating:** 2
**Confidence:** 3

**Summary:**

The paper addresses construction of deep spectral kernels. The paper first includes a theoretical part in which a few sin/cos-based feature map families are considered. It is argued that applying to them suitable stacking maps produces positive definite kernels with expanding RKHS. Then, a related generative spectral kernel framework is introduced. Finally, several sets of experiments with the proposed method are provided.

**Strengths:**

The results presented in the paper appear to be original.

Detailed derivations of theoretical results are provided in the appendix.

The reported results of experiments suggest a good performance of the proposed method compared against several alternative spectral kernel methods (but the code does not seem to be provided, so these results are not verifiable).

**Weaknesses:**

Clarity
--------

**Poor mathematical exposition.** I found the paper quite hard to read. From the very beginning, the exposition suffers from various mathematical inaccuracies and confusing statements.

- line 98: Yaglom's theorem is not carefully formulated. Theorem 2 states that a kernel is *positive definite* if and only if it is the Fourier transform of a *positive semi-definite bounded variation spectral density of a Lebesgue-Stiltjes measure*. To begin with, the paper never explains if "positive definite" means "strictly positive definite" or "positive semi-definite". Given that the semi-definiteness is explicitly mentioned, one would expect that positive definiteness is meant to be strict. However, the theorem is, of course, invalid in this case, because of the possibility $s(\\omega, \\omega)\\equiv 0$. Next, it is not explained what *bounded variation spectral density of a Lebesgue-Stiltjes measure* means precisely. The formulation of Bochner's (stationary) theorem is also not very careful: the non-negative measure $s(\\omega)d\\omega$ may have a singular component, not having a density.

- Moreover, it is not clear what is the point of stating Yaglom's theorem, since in the non-stationary case the spectral characterization is not transparent: the function $s(\\omega, \\omega)$ needs to be positive (semi-)definite like its Fourier image $k(x,x')$, so the description of kernels is not really simplified by Fourier transforming. (Of course, in the stationary case the situation is drastically different, since the spectral characterization in terms of non-negative measures is very transparent.)

- **(Most important)** The spectral function $s(\\omega, \\omega')$ appearing in Yaglom's theorem may be negative at $\\omega\\ne \\omega'$ - this does not contradict its positive definiteness. Positive definiteness of the function $s(\\omega, \\omega')$ and its positivity are two completely different notions (similarly to how the positive definiteness of a matrix is completely different from the positivity of its matrix elements). Then, I don't understand what is meant in definition 1 by the expectation $\\mathbb E_\\omega$ "*under the spectral distribution $s(\\omega, \\omega)$.*" In general, there is no probability distribution associated with the spectral function $s(\\omega, \\omega)$ if it is just positive-definite.

- line 140: the lemma states that a "reproducing kernel" remains a reproducing kernel. It's unclear what would be a "non-reproducing kernel". It appears that the authors mean positive (semi-)definite kernels.

- line 317: "*The generated $s(\\omega,\\omega')$ is a probability density function, satisfying the positive definiteness condition
of the spectral density in Theorem 2.*" - Again, it looks like the authors confuse the positive definiteness and positivity of $s(\\omega,\\omega')$.

**Poor motivation.** In addition to mathematical issues, the paper does not properly motivate the specific addressed questions, e.g. why we should be interested in the iterated sine/cosine maps and "removal" and "concatenations" scenarios as in eqs. (3)-(4)).

Contribution
----------------
I find the contribution of the paper to be very limited. The theoretical results such as Propositions 1 and 2 (assuming their correctness) are narrow and technical. Conclusions regarding the preservation of positive-definiteness or progressive expansion under stacking are more or less straightforward consequences of classical general theorems.

Regarding the experimental part, the authors don't seem to have open-sourced their code for independent verification.

**Questions:**

This paper is poorly written and certainly requires a major revision.

---

> ### Author Response · Authors · 2025-11-21
> **Response to Reviewer Ap4m**
>
> **W1, W2:** To ensure mathematical accuracy, we rewrite Section 2.1 as follows and have included it in the revised manuscript.
>
> **Theorem 1 (Bochner’s Theorem)** A continuous and stationary kernel $k(\boldsymbol{x}, \boldsymbol{x}') = k(\boldsymbol{\tau}), \boldsymbol{\tau} = \boldsymbol{x}- \boldsymbol{x}'$ on $\mathbb{R}^d$ is positive definite if and only if it can be formulated as: $k(\boldsymbol{\tau}) = \int_{\mathbb{R}^d} e^{i\boldsymbol{\omega}^\top \boldsymbol{\tau}} d \mu (\boldsymbol{\omega})$, where $\mu$ is a bounded non-negative measure on $\mathbb{R}^d$.
>
> Based on Bocher's theorem, if $\mu$ is absolutely continuous with respect to the Lebesgue measure, *i.e.,* $d\mu(\boldsymbol{\omega}) = s(\boldsymbol{\omega})d\boldsymbol{\omega}$, there exists a one-to-one correspondence between the kernel $k(\boldsymbol{\tau})$ and the spectral density $s(\boldsymbol{\omega})$. such that:
>
> $$
>  k(\boldsymbol{\tau}) = \int_{\mathbb{R}^d} s(\boldsymbol{\omega})e^{i\boldsymbol{\omega}^\top \boldsymbol{\tau}}d\boldsymbol{\omega}, \quad
>             s(\boldsymbol{\omega}) = \int_{\mathbb{R}^d} k(\boldsymbol{\tau}) e^{-i\boldsymbol{\omega}^\top \boldsymbol{\tau}}d(\boldsymbol{\tau}).
> $$
>
> **Theorem 2 (Yaglom's Theorem)** A continuous kernel $k(\boldsymbol{x}, \boldsymbol{x}')$ is positive semi-definite if and only if it can be formulated as: $k(\boldsymbol{x},\boldsymbol{x}')=\int_{\mathbb{R}^d \times \mathbb{R}^d}e^{i(\boldsymbol{\omega}^\top \boldsymbol{x} - \boldsymbol{\omega}'^\top \boldsymbol{x}')} \mu(d\boldsymbol{\omega},d\boldsymbol{\omega}')$, where $\mu$ is a bounded Hermitian positive semi-definite measure on $\mathbb{R}^d \times \mathbb{R}^d$.
>
> In particular, if $\mu$ is the Lebesgue-Stieltjes measure associated with some positive semi-definite spectral density function $s(\boldsymbol{\omega}, \boldsymbol{\omega}')$ with bounded variation *i.e.,* $\int_{\mathbb{R}^d \times \mathbb{R}^d} s(\boldsymbol{\omega}, \boldsymbol{\omega}') < \infty$ and satisfies $\mu(d\boldsymbol{\omega},d\boldsymbol{\omega}') = s(\boldsymbol{\omega},\boldsymbol{\omega}') d\boldsymbol{\omega}d\boldsymbol{\omega}'$, there exists a one-to-one correspondence between the positive semi-definite non-stationary kernel $k(\boldsymbol{x}, \boldsymbol{x}')$ and $s(\boldsymbol{\omega}, \boldsymbol{\omega}')$. Such that:
>
> $$
> k(\boldsymbol{x},\boldsymbol{x}')=\int_{\mathbb{R}^d \times \mathbb{R}^d}e^{i(\boldsymbol{\omega}^\top \boldsymbol{x} - \boldsymbol{\omega}'^\top \boldsymbol{x}')} s(\boldsymbol{\omega}, \boldsymbol{\omega}') d\boldsymbol{\omega} d\boldsymbol{\omega}',
> s(\boldsymbol{\omega}, \boldsymbol{\omega}') = \int_{\mathbb{R}^d \times \mathbb{R}^d}e^{-i(\boldsymbol{\omega}^\top \boldsymbol{x} - \boldsymbol{\omega}'^\top \boldsymbol{x}')}k(\boldsymbol{x},\boldsymbol{x}')d\boldsymbol{x} d\boldsymbol{x}'.
> $$
>
>
> **W3, W5**
>
> **Lemma 1** Let $\mu$ and $P$ be the a Lebesgue-Stieltjes measure and a probability measure on $\mathbb{R}^d \times \mathbb{R}^d$, respectively, with right-continuous monotonically increasing distribution functions $F_{\mu}:\mathbb{R}^d \times \mathbb{R}^d \to \mathbb{R}$ and $F_P:\mathbb{R}^d \times \mathbb{R}^d \to [0,1]$. The following equality holds:
>
> $$
> \mu(a, b] = F_{\mu}(b)-F_{\mu}(a) = C_{\mu}[F_P(b)-F_P(a)]= C_{\mu}P(a,b],
> $$
> where $C_{\mu} = \mu(\mathbb{R}^d \times \mathbb{R}^d)$ is a non-negative constant representing the value of $\mu$ on the universal set.
>
> This Lemma indicates that a corresponding probability density can be identified and applied to the non-stationary case, providing a way for computing an explicit spectral kernel mapping and subsequently constructing a deep spectral kernel.

---

> > ### Author Response · Authors · 2025-11-21
> > **Response to Reviewer Ap4m**
> >
> > **W4:** Yes, we indeed do not find the definition of non-reproducing kernels. Therefore, we have replaced “reproducing kernel” with “positive semi-definite kernel”.
> >
> > **Motivation and Contribution:**
> >
> > **Motivation:** Deep spectral kernels are constructed by hierarchically stacking explicit spectral kernel mappings. The explicit kernel mapping is commonly obtained through the kernel approximation using a sampling strategy based on the Fourier transform form of kernels. This makes the explicit kernel mapping defined in the complex domain with sine and cosine maps. To obtain a real-valued kernel, researchers tend to remove or concatenate the imaginary part. Thereby, we focus on the sine/cosine and "removal" and "concatenation" scenarios.
> >
> > In addition to mathematical issues, we also propose GensKer to address the limitations of existing deep spectral kernel methods: (1) the optimization challenge. By stacking the periodic functions (*i.e.,* the spectral kernel mappings), the deep spectral kernel is prone to local minima. (2) The range of spectral density. In most existing deep spectral kernel models, the spectral density function is data-independent and consists of predetermined components, restricting the range of spectral density even within deep architectures.
> >
> > **Contribution:** In this work, we first explore the general characterization of them under various frameworks (Equation (4) of the original manuscript). Furthermore, we propose GensKer to address the limitations of existing deep spectral kernel methods: the optimization challenge and the range of spectral density. This is presented in Section 4. Notably, the theoretical results guide the design of our GensKer architecture.

---

### Author Response · Authors · 2025-12-03
**Response Summary**

To address the limitations of most existing deep spectral kernel learning models: (1) the optimization challenge and (2) the range of spectral density, we propose GensKer, a framework designed to adaptively learn a data-dependent spectral density, from which the frequency matrix is subsequently produced to construct spectral kernels with single-layer spectral kernel mapping. Notably, the architecture of GensKer is guided by our theoretical results. Therefore, in this paper, we first investigate the general characterization of deep spectral kernels and identify favorable spectral kernel structures. Building on these results, we propose our GensKer model.

We sincerely thank reviewers for their careful review of our paper. The proposed insightful comments and suggestions are helpful in improving our work. We have posted detailed responses to each comment and have revised the manuscript accordingly. The major revisions are summarized as follows:

- Regarding the preliminary, we have carefully formulated the related theorems (Bochner's Theorem and Yaglom's Theorem) and highlighted the case (the newly included Lemma 1) discussed in our work. Furthermore, we have included the related fundamental concepts (spectral kernel).
- Regarding the theoretical analysis, we have included the more detailed analysis process (why treat $\mathcal{C}_ {s, w/o}^{L-1}$, $\mathcal{C}_ {ns, w/o}^{L-1}$, $\mathcal{C}_ {s, c}^{L-1}$, and $\mathcal{C}_ {ns, c}^{L-1}$ as constants). In addition, we have provided the formal statement of the theoretical analysis results ($\mathcal{H}_ {k^{L-1}} \subset \mathcal{H}_ {k^L}$).

- Regarding the proposed method, we have more clearly stated the motivation behind our proposed GensKer, including the specific addressed questions and the architectural design.

- Regarding the experiment, we have added a more detailed experimental setting and additional comparison (the statistical significance, the computational cost, and the baseline method).

- In addition, we have included more discussion regarding the limitations, future work, and related work of our paper.

---

### Meta-Review · Area_Chair_QZZu · 2026-01-07

**Summary:**

This paper proposes GensKer, a generative spectral kernel approach learns an adaptive spectral distribution with a generative network. Reviewers’ main objections are unclear exposition and questionable soundness (confusing/inaccurate preliminaries, unclear definitions/motivation), plus insufficient experimental/reproducibility details, thought part of the confusion might come from reviews' lack of familiarity to the topic; while the rebuttal adds clarifications and more experiment tables, these issues were not convincingly resolved.

In addition, the related work Implicit Kernel Learning (Li at al, AISTATS, 219) already learns the kernel spectral distribution via a deep generative model with random-feature sampling, which overlaps strongly with the proposed “generative spectral distribution” component and weakens novelty unless clearly differentiated.  We therefore reject due to remaining clarity/soundness concerns and weakened novelty positioning.

**Reviewer Concerns:**

Resolved: Most clarification questions are resolved, including definitions, preliminaries, constants in the statement and experiment details.

Outstanding: readability,  novelty  and reproducibility.

**Reviewer Scores:**

Ap4m, HXp9 and RCjB may raise the score by 1 as their concerns about clarification are addressed as mentioned in the above section.

---

### Decision · Program_Chairs · 2026-01-26

Reject